# Origin and differentiation trajectories of fibroblastic reticular cells in the splenic white pulp

Hung-Wei Cheng[1], Lucas Onder[1], Mario Novkovic[1], Charlotte Soneson [2], Mechthild Lütge[1], Natalia Pikor[1], Elke Scandella[1], Mark D. Robinson [2], Jun-ichi Miyazaki[3], Anne Tersteegen [4], Ursula Sorg[4], Klaus Pfeffer[4], Thomas Rülicke[5], Thomas Hehlgans[6] & Burkhard Ludewig [1]

The splenic white pulp is underpinned by poorly characterized stromal cells that demarcate distinct immune cell microenvironments. Here we establish fibroblastic reticular cell (FRC)-specific fate-mapping in mice to define their embryonic origin and differentiation trajectories. Our data show that all reticular cell subsets descend from multipotent progenitors emerging at embryonic day 19.5 from periarterial progenitors. Commitment of FRC progenitors is concluded during the first week of postnatal life through occupation of niches along developing central arterioles. Single cell transcriptomic analysis facilitated deconvolution of FRC differentiation trajectories and indicated that perivascular reticular cells function both as adult lymphoid organizer cells and mural cell progenitors. The lymphotoxin-β receptor-independent sustenance of postnatal progenitor stemness unveils that systemic immune surveillance in the splenic white pulp is governed through subset specification of reticular cells from a multipotent periarterial progenitor cell. In sum, the finding that discrete signaling events in perivascular niches determine the differentiation trajectories of reticular cell networks explains the development of distinct microenvironmental niches in secondary and tertiary lymphoid tissues that are crucial for the induction and regulation of innate and adaptive immune processes.

[1] Institute of Immunobiology, Kantonsspital St. Gallen, 9007 St. Gallen, Switzerland. [2] Institute of Molecular Life Sciences and SIB Swiss Institute of Bioinformatics, University of Zurich, 8057 Zurich, Switzerland. [3] Division of Stem Cell Regulation Research, Osaka University Medical School, 565-0871 Osaka, Japan. [4] Institute of Medical Microbiology and Hospital Hygiene, University of Düsseldorf, 40225 Düsseldorf, Germany. [5] Institute of Laboratory Animal Science, University of Veterinary Medicine Vienna, 1210 Vienna, Austria. [6] Institute of Immunology, Regensburg Center for Interventional Immunology (RCI) and University Medical Center of Regensburg, 93053 Regensburg, Germany. Correspondence and requests for materials should be addressed to B.L. (email: burkhard.ludewig@kssg.ch)

The mammalian spleen is the largest secondary lymphoid organ (SLO) that contains specialized microenvironments for hematopoiesis and control of erythrocyte turnover in the red pulp, while immune protection against blood-borne pathogens is secured in the white pulp[1,2]. Asplenic individuals and patients undergoing splenectomy bear a life-long risk of overwhelming infections[3] and are predisposed to succumb to septic shock[4]. The first line of immune defense is provided by the marginal zone of the white pulp where microbial antigens are captured by myeloid cells[5,6] and innate immune responses are initiated[7]. Both T- and B-cell compartments of the white pulp are underpinned by specialized fibroblastic stromal cells that provide a physical scaffold and generate chemokines and cytokines to facilitate efficient interaction between immune cells[8,9]. Selective loss of white pulp reticular cells, for example during viral infection, precipitates profound immunodeficiency[10] and immune functionality is restored only once reticular cell networks have been rebuilt[11].

Generation of functional immune environments in lymph nodes depends on maturation and functional specialization of fibroblastic reticular cells (FRC) that are associated with the broad expression of the mucin-type transmembrane protein podoplanin (PDPN)[12–14]. In the spleen, however, PDPN expression is expressed mainly by T-cell zone reticular cells (TRC), which produce the homeostatic chemokines CCL19 and CCL21[15]. Splenic B-cell zone reticular cells, as their counterparts in lymph nodes, express the chemokine CXCL13[15,16] and encompass the follicular dendritic cell (FDC) fraction, which retain antigen on their surface through the expression of complement receptors[17]. The marginal zone in human and murine spleens is underpinned by marginal reticular cells (MRC), which express the adhesion molecule MAdCAM-1 and CXCL13[18,19], and thereby foster interaction with innate lymphoid cells and B cells during the initiation of antibody responses[19]. Interestingly, while lymph node reticular cell subsets have been characterized extensively[20,21], molecular details on splenic white pulp reticular cell lineages have remained elusive.

The analysis of cellular lineage relationship requires knowledge on the embryonic origin and subsequent commitment steps that determine the critical nodes in the differentiation trajectories. Both red and white pulp fibroblastic stromal cells descend from stem cells in the embryonic splenopancreatic mesenchyme[2]. Early Nkx2-5 and Islet1-positive mesenchymal progenitors that appear at embryonic day (E) 10.5 have been shown to participate in the generation of white pulp reticular cell subsets[18]. Notably, splenic Nkx2–5+Islet1+ progenitors contribute as well to the fibroblastic stromal cell pool in the red pulp[18]. Moreover, the finding that cardiac Nkx2–5+ progenitors give rise to both smooth muscle cells and cardiomyocytes[22] indicates that Nkx2–5 expression occurs as one of the first differentiation steps of various mesenchymal cell populations. Hence, it is conceivable that the critical process of white pulp reticular cell progenitor commitment ensues during later stages of spleen development and that these cells may function as the main drivers for the structural organization of the white pulp.

Since the nature and the habitat of committed splenic white pulp mesenchymal lymphoid tissue organizer (mLTo) cells during embryonic development and in the adult have remained largely unknown, we combine single-cell genetic and molecular approaches to record the differentiation trajectories of white pulp reticular cells. The combination of in vivo cell-fate mapping and single-cell RNA-seq (scRNA-seq) analysis reveals that differentiation of white pulp reticular cell networks is dependent on lymphotoxin-β receptor (LTβR) signaling, while mural cell specification and sustenance of multipotent adult progenitor cells follow LTβR-independent trajectories. The close linkage of FRC and mural cell development suggests that distinct micro-environmental niches in secondary and tertiary lymphoid organs develop in a hierarchical process with LTβR signaling serving as a main switch and other—only partially known—secondary signals driving FRC subset specification.

## Results

**Reticular cell subsets in the splenic white pulp.** The *Ccl19* promoter is well-suited to genetically target the main subsets of FRC in lymph nodes[13,16,23,24] and Peyer's patches[25]. In the spleen, the Ccl19–Cre transgene highlights fibroblastic stromal cells in the T- and B-cell zones of the white pulp (Supplementary Fig. 1a) highlighting T-cell zone reticular cells expressing CCL21 (Supplementary Fig. 1b) and CXCL13-producing reticular cells in the B-cell zone (Supplementary Fig. 1c). Since constitutive Cre recombinase expression is not suitable to pinpoint cellular progenitor–progeny relationship, we generated an inducible cell-fate mapping system based on the expression of the tetracyclin transactivator (tTA) in *Ccl19*-expressing cells. The resulting Ccl19-tTA strain was crossed with LC-1[26] and R26R-EYFP mice to permit timely regulated Cre expression in the triple-transgenic mouse model termed Ccl19-iEYFP (Fig. 1a). In the spleen, the Ccl19-iEYFP transgene targeted fibroblastic stromal cells throughout the white pulp (Fig. 1b and Supplementary Fig. 1d), recapitulating the phenotype of the Ccl19-Cre mouse model (Supplementary Fig. 1a–c). Moreover, the expression of the red-fluorescent protein tdTomato (tdTom) monitors current *Ccl19* promoter activity in the T-cell zone and the marginal zone (Fig. 1c and Supplementary Fig. 1d). Pre- and postnatal exposure to doxycycline (Dox) completely blocked Cre recombinase-dependent EYFP expression, while production of the tdTom was not affected (Supplementary Fig. 1e) indicating that Cre recombinase activity can be stringently regulated in this model system. High-resolution confocal microscopy facilitated the distinction of the main EYFP+ reticular cell subsets with MRC in the marginal zone expressing MAdCAM-1 and CD157, FDC in the B-cell follicle that exhibit complement receptor 1 and 2 (CD35 and CD21, respectively) and CD157 expression, TRC with PDPN and αSMA expression and stem cell antigen-1 (Sca-1)-positive cells forming a perivascular reticular cell (PRC) network (Fig. 1d).

Expression of the LTβR in splenic reticular cells is essential for the formation of the splenic white pulp[27,28]. As expected, cell type-specific *Ltbr*-deficiency in Ccl19-iEYFP *Ltbr*fl/fl mice led to the failure to form the white pulp with the loss of specific white pulp microenvironments (Fig. 1e). Interestingly, EYFP+ cells were still detectable in spleens of conditionally *Ltbr*-deficient mice and were identified as αSMA-expressing myofibroblasts surrounding the splenic arterioles (Fig. 1f), suggesting that splenic white pulp formation is abolished when periarterial myofibroblasts fail to respond to lymphotoxin. Next, we used flow cytometric analysis to quantify the effect of cell type-specific *Ltbr*-deficiency on white pulp reticular cell development. EYFP+ reticular cells constituted only a minor fraction of all non-endothelial splenic stromal cells, which could be separated in PDPN+ TRC, CD35+ FDC, and MAdCAM-1+ MRC (Fig. 1g and Supplementary Fig. 1f). Moreover, Sca-1+ cells co-expressed PDGFRα (Fig. 1g), a trait that is characteristic for mesenchymal progenitor cells in the bone marrow[29–31]. Since CD157 was expressed by TRC, MRC, and FDC (Fig. 1g), co-staining for PDPN and CD157 was used to distinguish Sca-1+ PRC from TRC/MRC/FDC subsets leaving a fraction that is termed here as triple-negative cells (TNC) (Fig. 1g). Strikingly, the lack of *Ltbr* expression in EYFP+ cells led to an almost complete loss of MRC, TRC and FDC, while the proportion of PRC was significantly elevated (Fig. 1g, h). The fraction of TNC was not affected by the

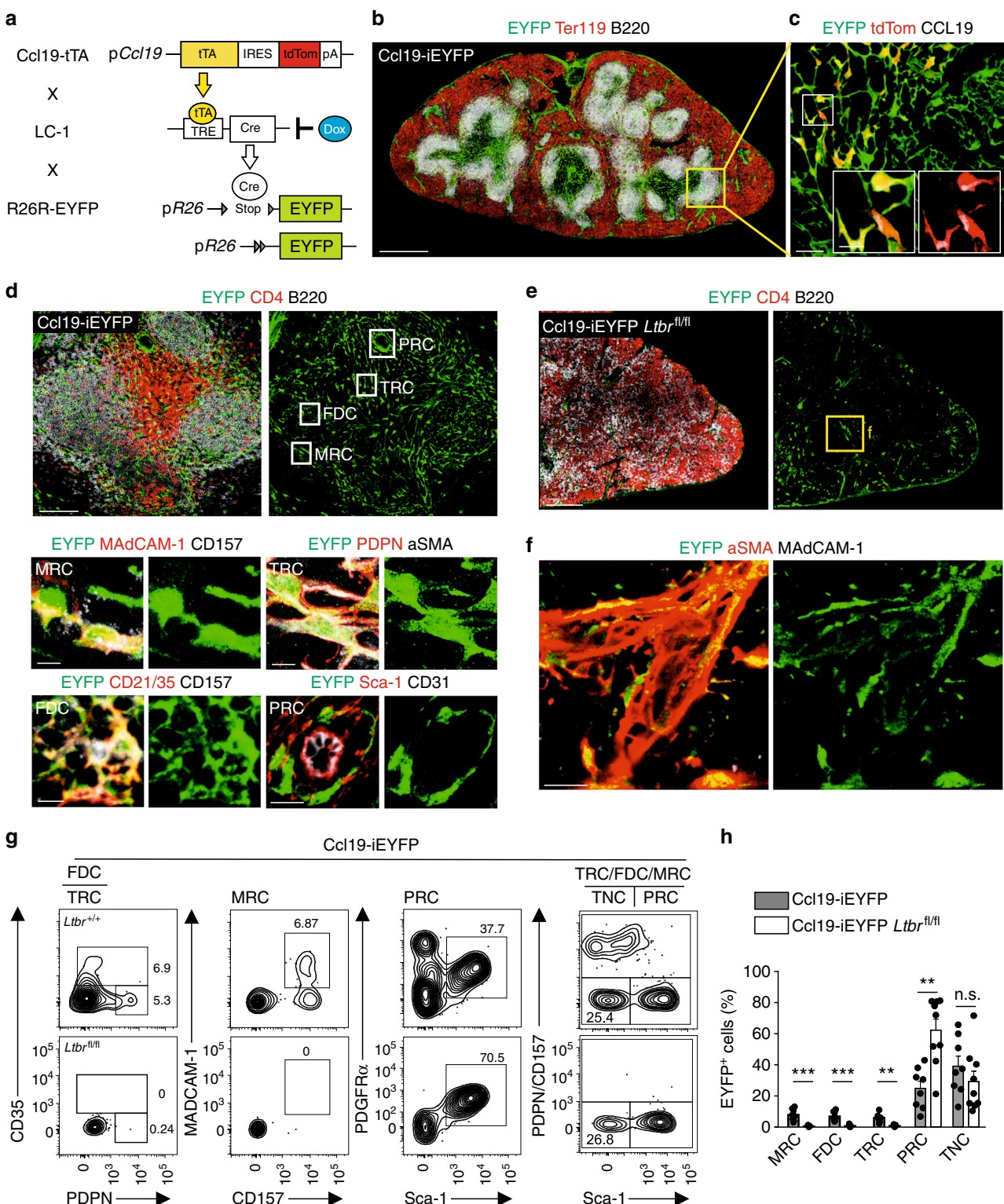

cell type-specific lack of *Ltbr* expression (Fig. 1g, h). In sum, these results indicate that mLTo cells are specifically targeted in the Ccl19-iEYFP transgenic mice and suggest that the LTβR-dependent emergence of distinct reticular cell subsets facilitates the organization of the splenic white pulp.

**Defining the mLTo cell habitat in the embryonic spleen.** Timed pregnancy and assessment of transgene activity at different embryonic stages in Ccl19-iEYFP mice revealed that αSMA-expressing myofibroblasts surrounding the splenic artery started to express the transgene at E19.5 concomitantly with the immigration of B and T cells to the fetal spleen (Fig. 2a and Supplementary Fig. 2a and b). At the neonatal stage (P0), expanded peri-arteriolar lymphocytic cuffs were underpinned by EYFP⁺ cells (Fig. 2a and Supplementary Fig. 2a and b). In a first differentiation step, MAdCAM-1⁺ MRC appeared at postnatal day (P7), forming a primordial marginal zone (Supplementary Fig. 2c). The reticular cell differentiation markers PDPN and

**Fig. 1** Differentiation of splenic white pulp reticular cell populations. **a** Schematic depiction of the triple transgenic Ccl19-iEYFP mouse model. **b** Spleen cross-sections from adult Ccl19-iEYFP mice without doxycycline treatment analyzed by confocal microscopy after staining with the indicated antibodies (scale bar = 500 μm). Boxed areas in (**b**) indicate the splenic white pulp EYFP+ reticular cells analyzed by confocal microscopy after staining with the indicated antibodies in (**c**, **d**). Boxed areas in (**c**) show region of higher magnification in the images (Scale bar = 50 μm and 10 μm in the boxes). Boxed regions in (**d**) indicate marginal zone reticular cells (MRC), follicular dendritic cells (FDC), T-cell zone reticular cells (TRC), and perivascular reticular cells (PRC), shown at higher magnification in lower panels (scale bars = 5 μm/10 μm). **e** Spleen section from adult Ccl19-iEYFP *Ltbr*fl/fl mice without doxycycline treatment stained with the indicated antibodies and analyzed by confocal microscopy. Box indicates magnified areas in (**f**) (scale bar in (**e**) = 200 μm, scale bars in (**f**) = 20 μm). **g** Representative flow cytometric analysis of viable CD45-depleted cells from Ccl19-iEYFP and Ccl19-iEYFP *Ltbr*fl/fl spleens without doxycycline treatment, gated on CD45−TER119−CD31−EYFP+ cells and analyzed for the indicated markers. Values indicate percentage of the respective population. **h** Percentage of EYFP+ cells from spleens of the indicated mouse strains, based on gating in (**g**) (n = 7–8 mice per group from 2 to 3 independent experiments, mean ± SEM). Statistical analysis was performed using Mann–Whitney test for (**h**). Microscopy data are representative for two or more independent experiments (n = 6 mice). Source data are provided as Source Data file

CD21/35 became detectable at postnatal week 2 with the beginning of T-cell and B-cell zone segregation (Fig. 2b and Supplementary Fig. 2d). Multicolor flow cytometric analysis revealed that the majority of the perivascular myofibroblasts detected at E19.5 and P0 exhibit a PRC phenotype, while the proportion of reticular cells expressing the differentiation markers PDPN and CD157 (Fig. 2c, d) increased with the growth of the splenic white pulp until week 6 (Supplementary Fig. 2e). Moreover, assessment of Ki67 expression revealed that the expansion of the white pulp was accompanied by high-rate proliferation of EYFP+ cells (Fig. 2e, f), suggesting that splenic white pulp formation is linked to distinct proliferation and differentiation processes of reticular cells that are triggered by the commitment of periarterial myofibroblasts at E19.5. Moreover, these data resolve that cells targeted by the Ccl19-tTA transgene function as embryonic mLTo cells during white pulp development.

**Tracking the progeny of embryonic mLTo cells**. To assess whether embryonic mLTo cells exhibit multipotent differentiation potential, we permanently tagged the first transgene-positive mLTo cells by treating pregnant Ccl19-iEYFP and Ccl19-iEYFP *Ltbr*fl/fl dams at E19.5 with Dox and provided the compound to the lactating dam, and after weaning to the offspring, via drinking water up until week 6 (Fig. 3a). Both E19.5–6 week fate-mapped *Ltbr*-proficient (Fig. 3b) and *Ltbr*-deficient (Fig. 3c) reticular cells were found mainly in areas surrounding the main branch of the splenic artery. *Ltbr*-proficiency facilitated differentiation of embryonic mLTo cells into PDPN+ TRC (Fig. 3d), MAdCAM-1+ MRC (Fig. 3d) and CD21/35+ FDC (Fig. 3e). Moreover, fate-mapped cells exhibited additional functional traits of FRC subsets such as CCL19 expression in TRC and CXCL13 expression in FDC (Supplementary Fig. 3a, b). In contrast, neither of the specialized reticular cell subsets appeared in spleens of fate-mapped Ccl19-iEYFP *Ltbr*fl/fl mice where *Ltbr*-deficient EYFP+ cells remained as αSMA-expressing myofibroblasts in their periarterial niches (Fig. 3f, g). Flow cytometric analysis confirmed the differentiation block in *Ltbr*-deficient fate-mapped mLTo cells (Fig. 3h and Supplementary Fig. 3c) and showed that *Ltbr*-dependent arrest of reticular cell development resulted in an accumulation of PRC (Fig. 3i and Supplementary Fig. 3c). Interestingly, the proliferation rates of EYFP+ fate-mapped cells determined as fraction of Ki67+ cells were not affected by the *Ltbr* ablation (Fig. 3j, k). Taken together, these data reveal that embryonic mLTo cells operate as multipotent reticular cell progenitors and indicate that the maintenance of the PRC network is independent of LTβR signaling.

To further investigate the progenitor-progeny relationship of embryonic mLTo cells and FRC subsets at the single-cell level, we crossed the Ccl19-tTA mouse strain to the LC-1 strain and the multicolor reporter Brainbow2.1 mouse line (Ccl19-iBrainbow2.1)[32]. As expected, FRC were labeled randomly by different

fluorescent proteins in the absence of doxycycline application (Supplementary Fig. 3d–e). In contrast, individual cell fate-mapping of E19.5 mLTo cells to 6 weeks revealed that single colored EYFP+ or RFP+ cell clusters contain distinct FRC subsets with PDPN+ TRC, CD21/35+ FDC or single MAdCAM1+ MRC, which is indicative of locally proliferating progenitors (Supplementary Fig. 3f–h). These results suggest that FRC subset specification from embryonic mLTo cells occurs through clonal expansion in local microenvironments during the interaction with hematopoietic cells.

**Consecutive mLTo cell commitment along the arteriolar tree**. The finding that progeny of E19.5 mLTo cells congregate mainly in the area surrounding the main splenic artery (Fig. 3b) suggested that the process of mLTo cell commitment starts at the central blood supply of the developing organ. Indeed, whole-mount scanning of Ccl19-iEYFP embryonic spleens revealed that arterial smooth muscle cells undergo a decisive commitment step at E19.5 along the main branch of the splenic artery (Supplementary Movie 1). Moreover, the finding that the white pulp structures developed in E19.5–6 week fate-mapped *Ltbr*-deficient mice (Fig. 3c) suggested that *Ltbr*-proficient mLTo cells could be committed after E19.5. To determine the spatio-temporal parameters underlying mLTo commitment during spleen development, we blocked Cre recombinase activity in Ccl19-iEYFP mice through Dox administration from E19.5 (Fig. 4a) or from P7 (Fig. 4b) to week 6 and compared the expansion of EYFP+ progeny to untreated mice. We found that mLTo cell progeny in distal regions were present at lower density (Fig. 4c), but still contributed to the MRC (Fig. 4c, arrow), TRC (Fig. 4c, arrowhead), and FDC (Fig. 4c, asterisk) subsets. White pulp areas more proximal to the central artery were densely populated with all reticular cell subsets (Fig. 4c). Arresting Cre recombinase activity at P7 was associated with equally dense cell distribution in distal and in proximal (Fig. 4d) areas. Next, we applied a histo-morphometric analysis that resolves the concentric distance of EYFP+ cells from the central artery at the hilum of the spleen (Fig. 4e). Quantification of the frequency of EYFP+ cells in concentric areas relative to the central artery revealed that density and distribution pattern of white pulp reticular cells were determined during the time period of E19.5 to P7 (Fig. 4f). Since fate mapping of potential mLTo cells at E17.5 did not yield any EYFP+ cells (Supplementary Fig. 3i), we conclude that mLTo cell commitment starts at E19.5 and continues along the growing arteriolar tree during the first week of postnatal growth.

**Transcriptome-based subset definition of splenic fibroblasts**. To resolve the molecular characteristics of white pulp reticular cells and to determine their differentiation trajectories, we first performed population-based RNA sequencing (popRNA-seq)

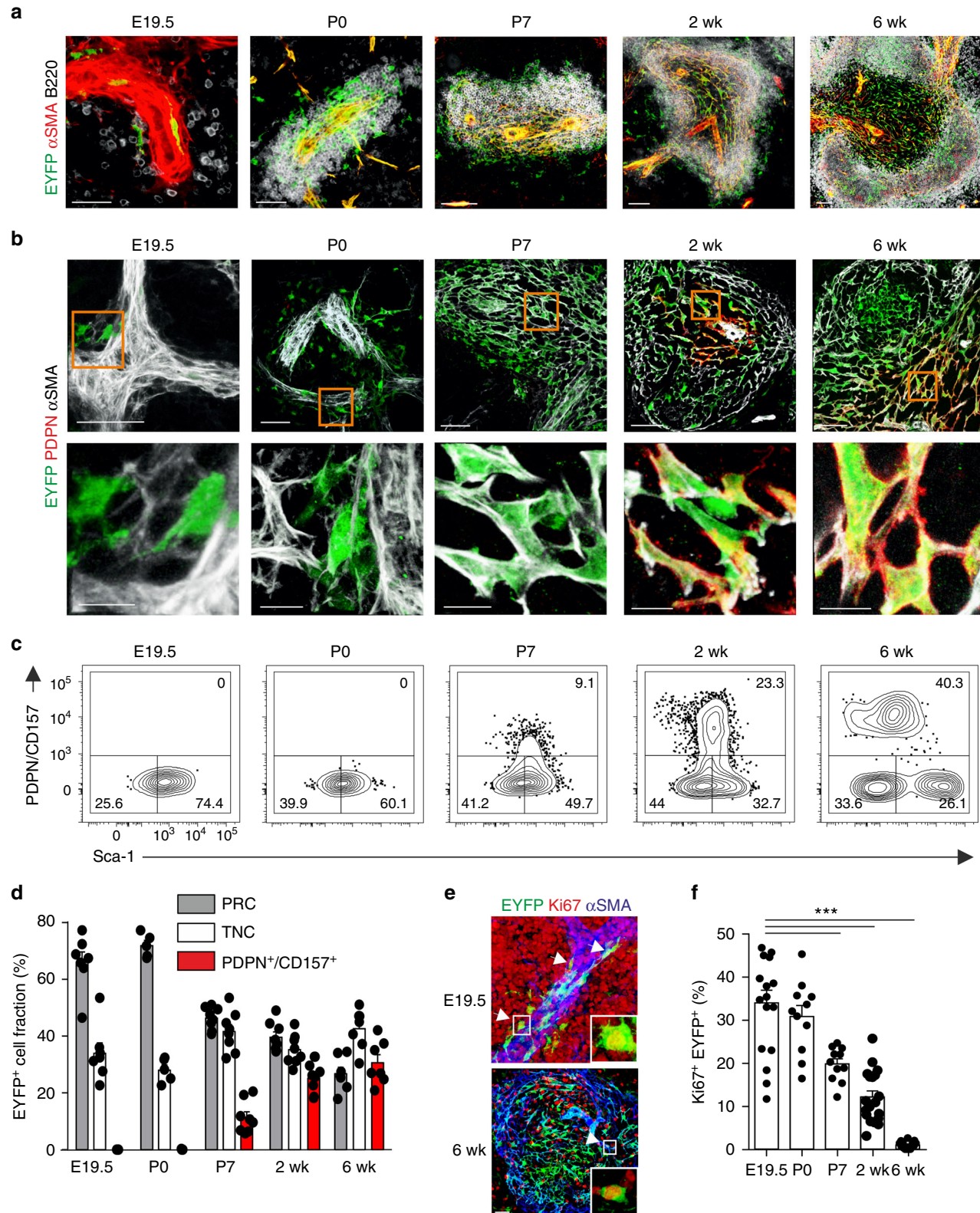

using the established cell marker profiles (Fig. 1g). We analyzed the transcriptome of sorted embryonic mLTo cells based on EYFP expression and adult *Ltbr*-deficient EYFP⁺ cells to elucidate the molecular basis of postnatal mLTo cell stemness. Data analysis based on differentially expressed genes revealed that the gene expression pattern of embryonic mLTo cells was distinct from adult reticular cells (Supplementary Fig. 4a). Gene set enrichment analysis indicated that embryonic periarterial mLTo cells differ from *Ltbr*-arrested adult PRC mainly in terms of antigen processing and presentation and the generation of extracellular matrix components (Supplementary Fig. 4b). Gene-expression analysis provided more detailed molecular insight and revealed that both embryonic mLTo cells and *Ltbr*-deficient EYFP⁺ cells expressed transcription factors implicated in spleen

**Fig. 2** Reticular cell subset differentiation during splenic white pulp development. **a, b** Time course analysis of EYFP-expressing cells in spleens from Ccl19-iEYFP mice harvested at the indicated stages. Confocal microscopic analysis depicting αSMA+ myofibroblasts (**a**) and PDPN+ TRC (**b**). Boxes in (**b**) indicate magnified regions in the lower row. Microscopy data are representative for two or more independent experiments (n ≥ 4 mice per group). **c, d** Flow cytometric analysis of CD45−TER119−EYFP+ cells using antibodies against PDPN, CD157 and Sca-1. Values from one representative experiment indicate percentage of the respective population shown in (**c**) and cumulated values for proportions of reticular cell subsets shown in (**d**) (n > 5 mice per group from two to three independent experiments, mean ± SEM). **e, f** Assessment of cellular proliferation using Ki67 staining in spleens of Ccl19-iEYFP mice at the indicated age with representative section in (**e**) and quantification of the proportion of Ki67 expression in EYFP+ cells (**f**) (n > 4 mice per group from three independent experiments, mean ± SEM). Statistical analysis was performed using a one-way ANOVA with Tukey's post test. Source data are provided as Source Data file

organogenesis as well as mesenchymal stem cell markers (Supplementary Fig. 4c). The reduced expression of TRC, MRC, and FDC marker genes in adult *Ltbr*-deficient EYFP+ cells (Supplementary Fig. 4d) suggested that initiation of TRC/MRC/FDC subset specification and concomitant loss of stemness are the major LTβR-dependent processes during white pulp formation. Indeed, comparison of popRNA-seq profiles of defined reticular subsets with E19.5 EYFP+ cells revealed that embryonic mLTo cells were most similar to PRC (Supplementary Fig. 4e). The two cell populations shared mesenchymal stem cell and splenic organogenesis marker expression, while molecular traits of TRC, MRC, and FDC were mostly absent in the PRC fraction (Supplementary Fig. 4f). Overall, these data support the notion that initiation of TRC/MRC/FDC subset specification and concomitant loss of stemness are the major LTβR-dependent processes during white pulp formation.

To further decipher the differentiation trajectories of splenic FRC subsets and to determine the heterogeneity of splenic reticular cells at the single-cell level, we performed two different scRNA-seq analyses (Fig. 5a). The global scRNA-seq analysis of splenic fibroblastic stromal cells revealed eleven distinct cell clusters overlaid on a t-distributed stochastic neighboring embedding (tSNE) plot (Fig. 5b) with six clusters harboring cells that exhibited EYFP mRNA expression (Fig. 5c, d). Almost all cells in cluster 7 showed high *Eng*, *Pdgfrb*, and *Ly6a* expression, but were largely EYFP mRNA-negative (Fig. 5d). Confocal microscopy analysis revealed that Endoglin protein expression mainly highlighted red pulp fibroblasts and PRC of central arteries in the white pulp (Supplementary Fig. 5a). Expression of *Cd34* was restricted to cells in clusters 8–11, with mRNA of the extracellular matrix protein Lumican (*Lum*) highlighting clusters 10 and 11 and the mRNA of the mesothelial cell marker Mesothelin (*Msln*) being expressed only in clusters 8 and 9 (Fig. 5d). We found that Lumican+ cells form a subcaspsular fibroblast layer (Supplementary Fig. 5b), while Mesothelin+ cells form the outer layer of the splenic capsule (Supplementary Fig. 5c). Importantly, EYFP+ cells were scarce in the subcapsular layer and completely absent in the mesothelial cell layer (Supplementary Fig. 5b, c). Utilizing the additional marker combinations derived from the global scRNA-seq analysis, we have elaborated a flow cytometry protocol that facilitates distinction of red pulp and white pulp fibroblastic stromal cells (Supplementary Fig. 5d). Overall, more than 80% of ICAM-1+ CD157+ Endoglin− white pulp fibroblasts expressed the EYFP marker (Supplementary Fig. 5d).

Next, we employed single-cell transcriptomic analysis on sorted EYFP+ cells from Ccl19-iEYFP spleens. As expected, the scRNA-seq analysis revealed six distinct clusters overlaid on a tSNE plot with PRC, TRC, MRC, and FDC subsets (Fig. 5e, f) that could be identified through the gene signatures obtained by popRNA-seq (Supplementary Fig. 4). In accordance with the flow cytometry-based subset definition (Fig. 1), the two remaining clusters were assigned as triple negative cells (TNC1 and TNC2, respectively) (Fig. 5e). The TNC1 population showed enriched

expression for canonical mural cell markers such as *Pdgfrb*, *Cnn1*, *Cspg4* (also known as NG2), and *Mcam* (CD146) (Fig. 5g), while the TNC2 population showed highly upregulated genes that are involved in cell cycling such as *Mki67*, *Mcm7*, *Cdk6*, and *Cks2* (Fig. 5h). Taken together, these transcriptome analyses unveil the heterogeneity of splenic white pulp FRC subsets and reveal the existence of additional cell populations that relate to the immune-interacting reticular cells.

**Differentiation trajectories of splenic FRC subsets.** To corroborate FRC subset identity and to elucidate further details in the FRC differentiation trajectories, we incorporated EYFP+ cells from adult Ccl19-iEYFP *Ltbr*fl/fl spleens into the scRNA-seq analysis. The tSNE plot from *Ltbr*-deficient EYFP+ cells confirmed the lack of TRC, MRC, and FDC differentiation and the accumulation of an extended PRC population, while both TNC populations were maintained (Supplementary Fig. 6a). Interestingly, an additional cell cluster appeared in *Ltbr*-deficient EYFP+ cells, designated as intermediate PRC (int-PRC) (Fig. 6a and Supplementary Fig. 6a). Int-PRC are characterized by the expression of PRC markers (e.g., *Ly6a*) and the lack of TRC/MRC/FDC differentiation markers (e.g., *Bst1*, *Enpp2*, and *Clu*) (Supplementary Fig. 6b). Deconvolution of the FRC differentiation trajectories was achieved by employing the Monocle routine[33] that indicated PRC as progenitors for all differentiated white pulp reticular cell subsets (Fig. 6b). The Monocle analysis revealed two main differentiation trajectories with the LTβR-dependent branch leading toward TRC/MRC/FDC clusters and a LTβR-independent branch with the TNC1 population, while int-PRC lined-up along both differentiation routes (Fig. 6b and Supplementary Fig. 6c). To further resolve the LTβR-dependent differentiation of TRC/MRC/FDC populations, we re-embedded the cells situated along the LTβR-dependent trajectory and re-employed the Monocle analysis. The Monocle trajectory of the LTβR-proficient cells demarcated the continued differentiation of PRC through the stage of int-PRC towards a first branching node (node 1) leading to a split into TRC vs. MRC/FDC populations (Fig. 6c). Along the MRC/FDC route, a second node (node 2) mainly separated MRC from FDC subsets, albeit the absence of a distinct branch. In conjunction with the cell fate-mapping experiments, these data indicate that the PRC fraction harbors postnatal mLTo cells that differentiate through an intermediate stage towards the TRC, FDC, and MRC in a LTβR-dependent fashion, while the maintenance of the peri-arterial PRC network is independent of this pathway.

To further decipher the relation of differentiated FRC subsets with LTβR-independent TNC subsets, we first screened the marker genes of PRC, int-PRC, and TNC1 populations based on the scRNA-seq analysis. Expression of mesenchymal stem cell markers such as *Ly6a* (Sca-1), *Eng* (CD105), *Vcam1*, and *Pdgfra*[31,34] was highly enriched in PRC and int-PRC clusters, while the mural cell markers *Mcam* (CD146), *Rgs5*, *Cspg4*, and *Cnn1* (Calponin)[35] were mainly enriched in the TNC1 cluster

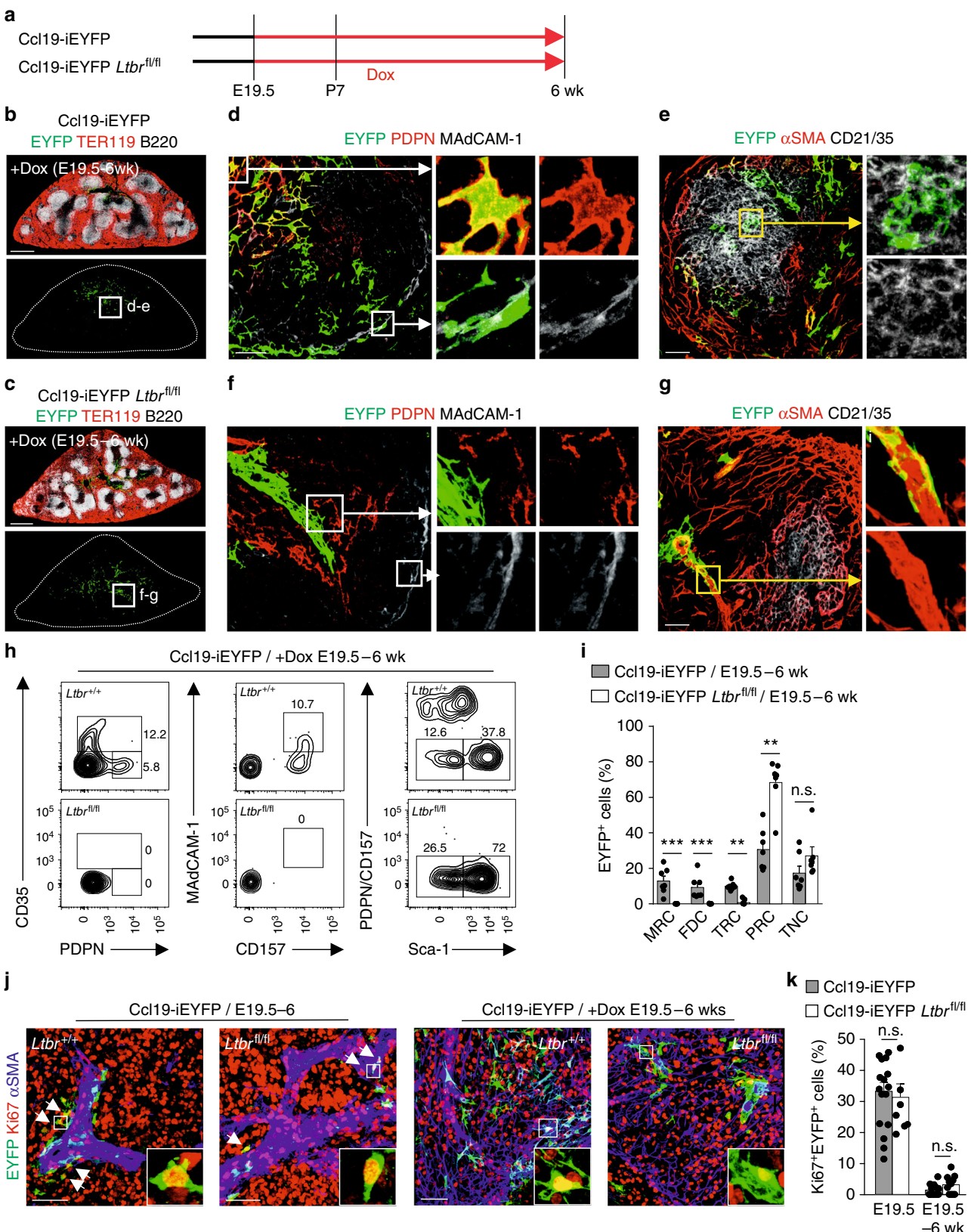

(Fig. 6d). Moreover, the expression of common perivascular cell markers *Itgb1* (CD29) and *Pdgfrb* was upregulated at similar levels in the PRC, int-PRC, and TNC1 populations (Fig. 6d) suggesting that TNC1 represent mural cells. Flow cytometric analysis of EYFP+CD157− reticular cells confirmed that TNC1 express CD29, but can be distinguished from PRC by the lack

of Sca-1 expression and the lack of CD44 and reduced CD106 expression (Fig. 6e). Confocal microscopy revealed that CD29+Calponin-1+ TNC1 form a distinct mural cell layer around white pulp arterioles (Fig. 6f). Gene-set enrichment analysis substantiated that this distinct mural cell subset contributes to blood vessel contractility and lacks the features

**Fig. 3** Cell fate mapping of embryonic lymphoid tissue organizer cells. **a** Schematic depiction of the experimental approach. Spleens have been harvested from E19.5 to 6 week fate mapped, **b** Ccl19-iEYFP, and **c** Ccl19-iEYFP *Ltbr*^fl/fl mice. Cross-sections were stained with the indicated antibodies and analyzed by confocal microscopy. Boxes indicate representative white pulp areas shown at higher magnification in (**d–g**) (scale bars = 500 μm). **d, e** Confocal microscopy analysis of fate-mapped *Ltbr*-proficient EYFP+ white pulp reticular cell subsets highlighting PDPN+ TRC (**d**), MAdCAM-1+ MRC (**d**) and CD21/35+ FDC (**e**), scale bars = 20 μm/5 μm. **f, g** Fate-mapped *Ltbr*-deficient EYFP+ cells by confocal microscopy using indicated marker combinations, scale bars = 20/5 μm. Microscopy data are representative for *n* = 7 mice per group from three independent experiments. **h** Representative flow cytometric analysis of fate-mapped EYFP+ cells from Ccl19-iEYFP and Ccl19-iEYFP *Ltbr*^fl/fl spleens, gated on and analyzed for the indicated markers. Values indicate percentage of respective populations. **i** Percentage of EYFP+ cells from spleens of indicated mouse strains, based on gating in (**h**) (*n* = 7 mice per group from three independent experiments, mean ± SEM). **j, k** Assessment of cellular proliferation using Ki67 staining in spleens of Ccl19-iEYFP and Ccl19-iEYFP *Ltbr*^fl/fl mice at the indicated age with representative section in (**j**) and quantification of the proportion of Ki67 expression in EYFP+ cells in (**k**) (*n* = 3–5 mice per group from three independent experiments, mean ± SEM). Statistical analyses were performed using a Mann–Whitney test. Source data are provided as Source Data file

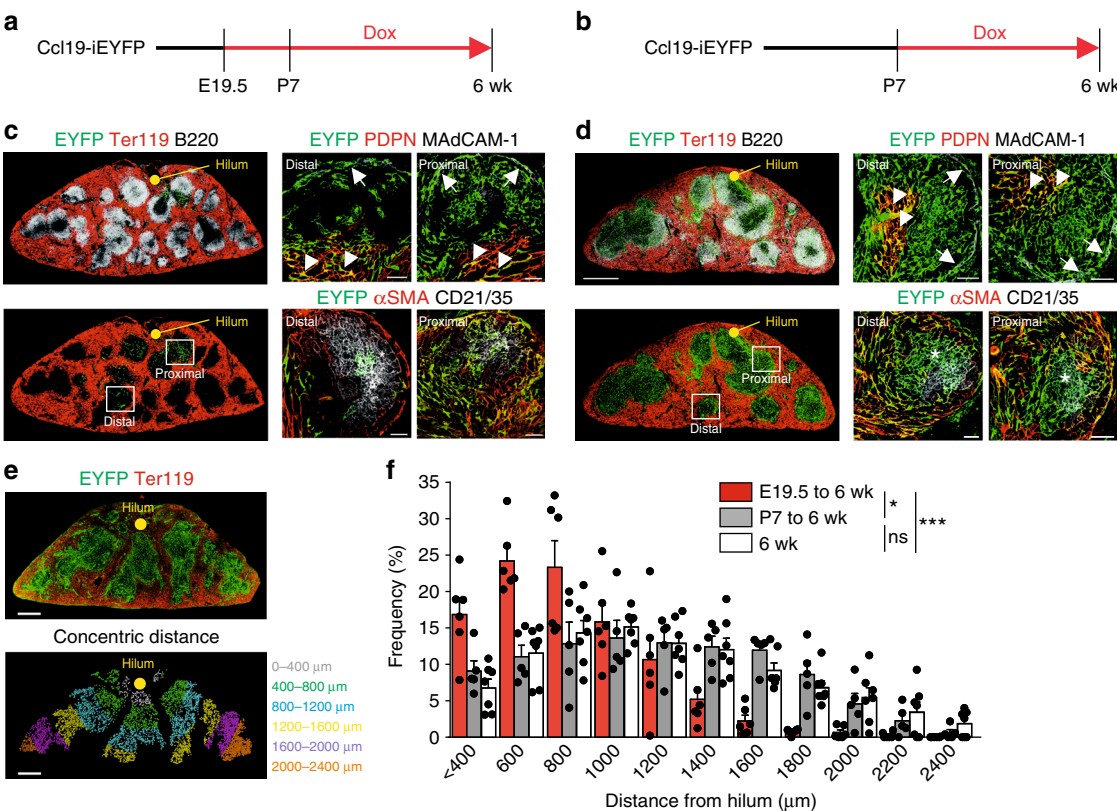

**Fig. 4** Staggered commitment of splenic white pulp organizer cells along the developing arterial tree. **a, b** Ccl19-iEYFP mice were treated with doxycycline from E19.5 (**a, c**) or P7 (**b, d**). Distribution and phenotype of EYFP+ cells in the white pulp of 6-week-old mice were determined by confocal microscopy using the indicated antibodies. Boxes show region of higher magnification in right images, arrows indicate EYFP+MAdCAM-1+ MRC **c, d**, arrowheads indicate EYFP+PDPN+ TRC (**c, d**), asterisks indicate EYFP+CD21/35+ FDC (**c, d**) (scale bars = 500/20 μm). **e** Determination of EYFP+ cell distances from the central artery at the hilum. Cross-sections from Ccl19-iEYFP spleens from 6-week-old mice were stained with antibody against TER119 and EYFP expression was pseudocolor-coded (lower image) according to concentric distance from the hilum (0–2400 μm). Scale bar = 250 μm. **f** Frequency of EYFP+ cells in white pulp areas of indicated concentric distances from the central artery (*n* = 5–7 mice per group from two independent experiments, mean ± SEM). Statistical analysis was performed using a Kolmogorov–Smirnov test. Source data are provided as Source Data file

and functions of the differentiated FRC subsets (Supplementary Fig. 6d). The TNC2 population was characterized by an enrichment of genes that are active during cell cycle and translation initiation (Supplementary Fig. 6d). Whether and how these proliferating EYFP+ cells give rise to LTβR-dependent or -independent fibroblastic stromal cell populations could not be resolved using the Monocle routine. In sum, our results indicate that adult PRC descend from embryonic periarterial mLTo cells and pass through an intermediate stage before developing into either mural cells or splitting up into the differentiation pathways of the main LTβR-dependent reticular cell subsets (Fig. 6g).

## Discussion

The current study has identified the embryonic origin and the differentiation trajectories of the adult reticular cell subsets that determine the structural organization of specialized immune environments in the splenic white pulp. Our findings reveal that stemness and multipotency are endowed to periarterial progenitor cells that contribute to the maintenance of the vascular structure and enable systemic immune surveillance in the splenic white pulp through subset specification of immune-interacting reticular cells.

Currently, competing views exist that promote divergent scenarios of reticular cell development in SLOs. Lymph node reticular

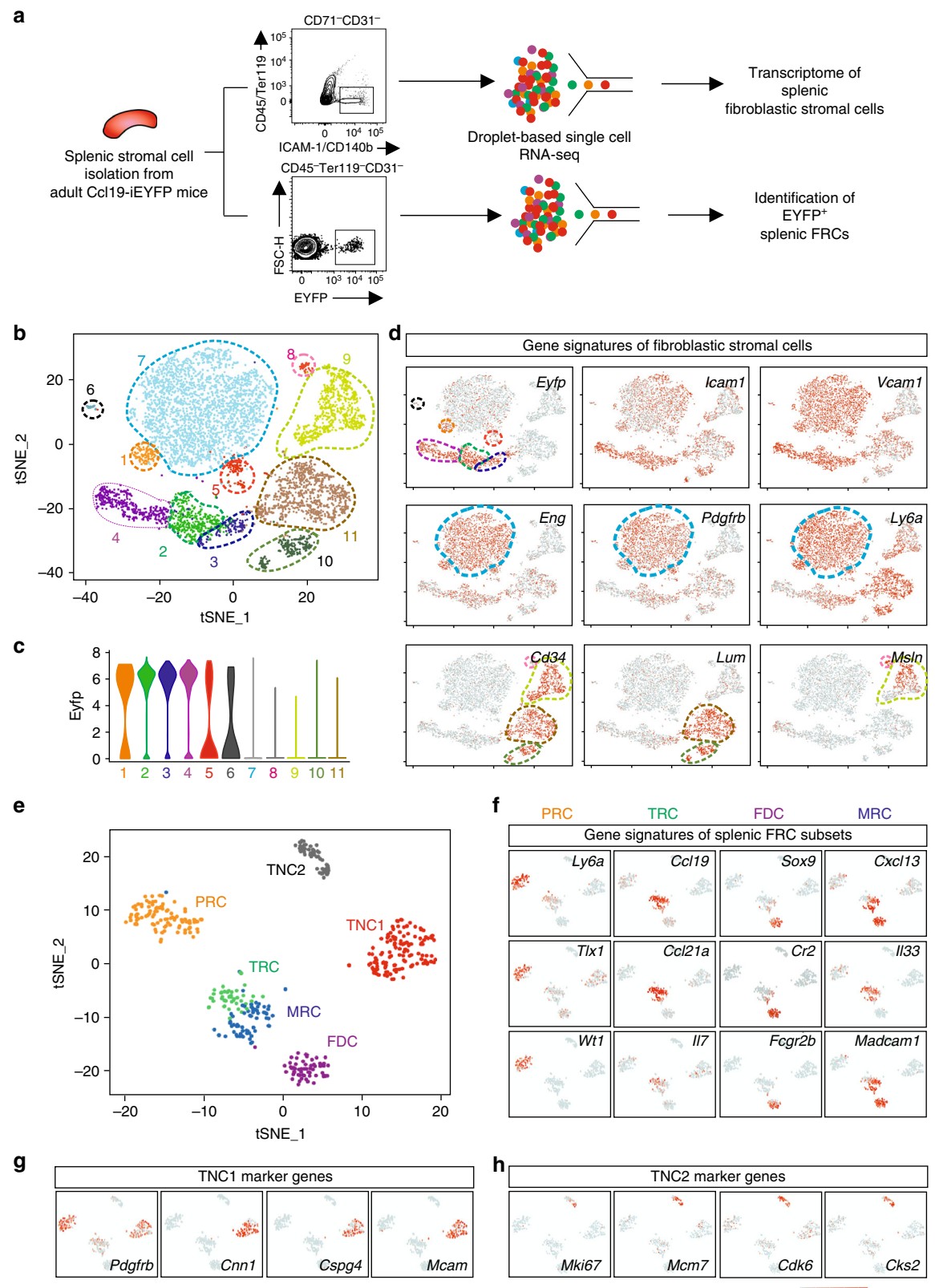

**Fig. 5** Transcriptomic analysis of splenic white pulp reticular cells. **a** Schematic depiction of the droplet-based scRNA-seq workflow. **b** tSNE map of sorted splenic fibroblastic stromal cells from adult Ccl19-iEYFP mice. **c** Violin plots of Eyfp expression in different clusters based on the scRNA-seq analysis. **d** Gene signatures of distinct splenic reticular cell subsets. The density of the red color represents the expression level of the genes. **e** tSNE map of sorted EYFP+ cells from adult Ccl19-iEYFP mice demarcated with the reticular cell populations. scRNA-seq data are pooled from two independent experiments. **f–h** Gene signatures of distinct splenic reticular cell subsets acquired from scRNA-seq analysis. The density of the red color represents the expression level of the genes

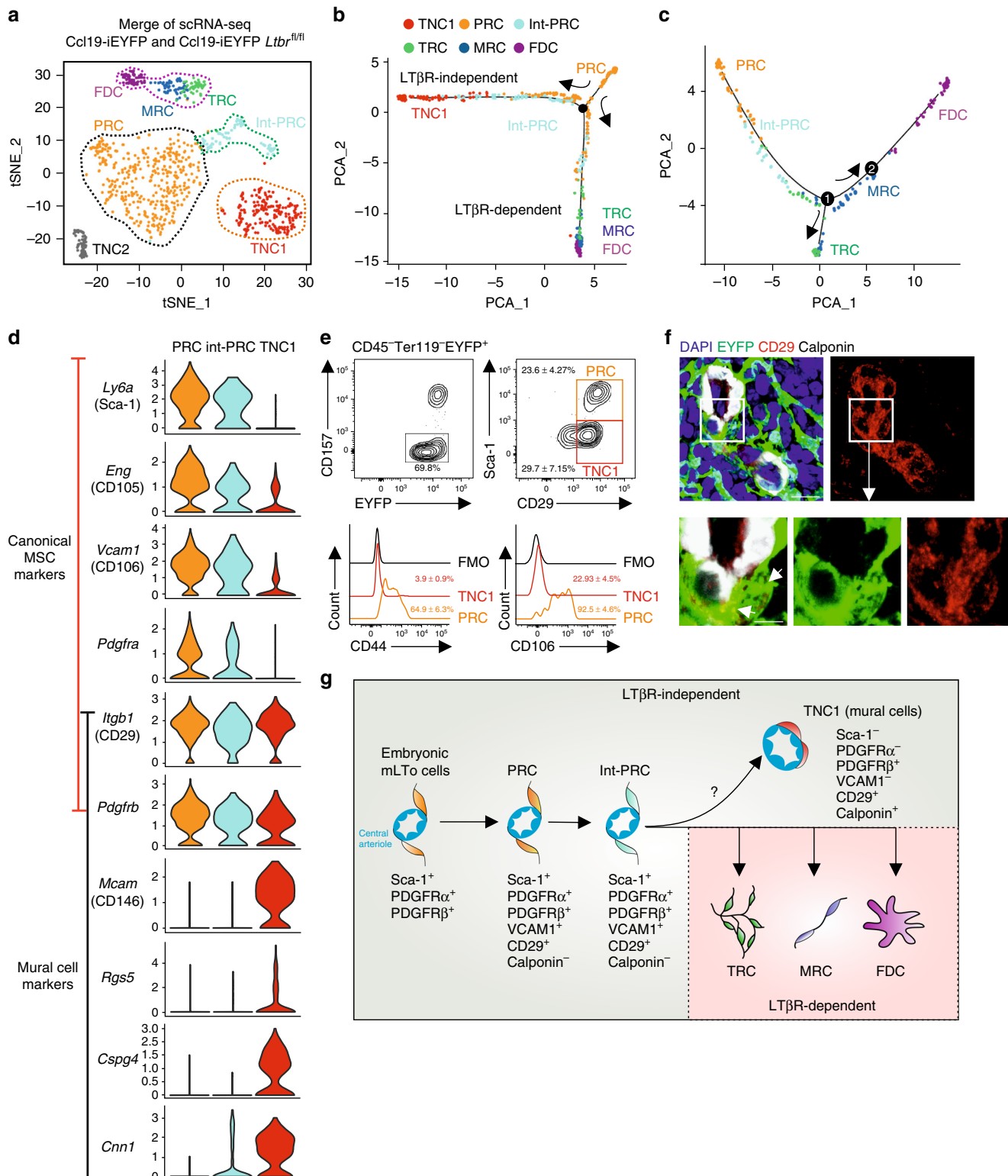

cells, for example, have been suggested to originate from pre-adipocyte progenitors[36], which are triggered by retinoic acid from adjacent neurons to produce CXCL13[37]. Based on the overlap in the expression of selected markers, MRC have been suggested to represent an intermediate mLTo cell[38] and to give rise to FDC[39]. Assessment of SLO development in globally gene-deficient animals combined with heterotopic transfer of stromal cell populations has led to the conclusion that adventitial progenitors from SLOs can give rise to one[40] or several reticular cell subsets[41]. Moreover,

an endothelial cell–MRC hybrid cell has been proposed to function as key organizer cell during postnatal spleen regeneration[42]. These profound conceptual discrepancies could be resolved here through the generation of a cell-fate mapping system that targets the committed progenitors of the main reticular cell populations. Importantly, the Ccl19-iEYFP model excludes tracing of irrelevant mesenchymal cell lineages such as red pulp fibroblasts[18] or myo-cardial smooth muscle cells[22]. Fate mapping of the relevant cell population has unveiled a discrete embryonic periarterial

**Fig. 6** Differentiation trajectories of reticular cell subsets in the splenic white pulp. **a** Merged tSNE cell map of sorted splenic EYFP+ cells from adult Ccl19-iEYFP and Ccl19-iEYFP *Ltbr*fl/fl spleen highlight the distinct reticular subsets. **b** Differentiation trajectory analysis of reticular cell populations constructed by the Monocle routine. Single cells are ordered along the trajectory tree according to the pseudo-time development. **c** Re-embedding of cell clusters along the LTβR-dependent trajectory re-analyzed with Monocle. Nodes indicate the bifurcating branching points of distinct cell populations along the differentiation trajectories. **d** Violin plots of canonical mesenchymal stem cell and mural cells markers expressed by PRC, int-PRC, and TNC1. **e** Representative flow cytometric analysis of EYFP+ cells from adult Ccl19-iEYFP spleen, gated on and analyzed for the indicated markers. Values indicate percentage of respective populations. (*n* = 6 mice per group from two independent experiments, mean ± SEM). **f** Spleen cross-sections from adult Ccl19-iEYFP mice analyzed by confocal microscopy after staining with the indicated antibodies (scale bar = 10 μm). Boxed areas in (**f**) indicate the magnified region below (scale bar = 5 μm). **g** Schematic of the LTβR-independent and -dependent differentiation trajectory proposed for embryonic mLTo cells and postnatal PRC, and differentiated reticular cell populations. Source data are provided as Source Data file

progenitor niche for all reticular cell subsets. Stemness and multipotency of the periarterial progenitors were maintained through adulthood providing a source for both FRC subsets and mural cells. The distinct, LTβR-dependent differentiation trajectories of TRC, MRC, and FDC suggest that these subsets integrate different sets of signals to generate suitable microenvironments of the diverse immune cell populations. Whether MRC represent a progenitor state[39] or a separate differentiation path could not be finally resolved in our analysis. Nevertheless, the Ccl19-iEYFP model will permit further dissection of reticular cell functions and differentiation pathways through cell-type specific and timed ablation of key molecules at different developmental stages and during ongoing immune responses.

Multipotent progenitor cells within the perivascular niche are usually referred to as mesenchymal stem cells[43] because these cells exhibit osteogenic, chondrogenic, and adipogenic potential[31,44] and contribute to pathologic fibrosis in multiple organs[45]. Perivascular stem cells are characterized by the expression of PDGFRα, PDGFRβ, and Sca-1[31,44], markers that are also expressed by PRC as shown in this study. The finding that the expression of the pericyte markers CD146 and NG2, encoded by *Mcam* and *Cspg4*, respectively, is associated with multipotent differentiation potential[44] has led to the conclusion that pericytes can function as mesenchymal stem cells. Recent studies, however, have challenged this view because even though pericytes exhibit tri-lineage differentiation in vitro, these properties could not be confirmed using in vivo cell-fate mapping[46]. Our finding that mural cells designated as TNC1 express the classical pericyte markers NG2, CD146, and RSG5[35], while the PRC and int-PRC were negative for these markers, support the conclusion that splenic reticular cells do not originate from pericytes. Clearly, splenic periarterial PRC possess multipotent stem cell functions and can be programmed to differentiate either into cells that promote vascular wall stability or give rise to different reticular cell subsets in a process that depends on signals provided in the respective microenvironment.

The evolution of SLOs provides important insight into how reticular cell subset specialization has followed developmental programs of lymphocytes. While specific antigen receptor-bearing cells appeared some 500 million years ago in a common vertebrate ancestor[47], lymph nodes with their fully developed set of functional microenvironments are a rather late amendment to the immune system which emerged ~100 million years ago in mammals[48]. In particular, the evolution of the spleen from amphibians to birds, starting ~300 million years ago, illustrates the adaptation of the white pulp infrastructure to the developing adaptive immune system[49]. Amphibians such as *Xenopus laevis* possess a distinguishable white pulp with lymphocytes forming concentric layers around the central arterioles and the white pulp in reptiles is underpinned by reticular cells immediately adjacent to the arteriole[49]. In subsequent evolutionary steps in birds, the splenic white pulp developed bursa-dependent germinal centers[50] that are characterized by the presence of FDC[51]. Hence, gain of more complex functions of lymphocytes, such as affinity

maturation in germinal centers, was accompanied by the co-evolution of the suitable microenvironments that are built by specific reticular cell niches. Such cellular interactions require hard-wired signaling circuits that permit on-demand formation of the suitable stromal cell infrastructure.

The molecular dissection of the reticular cell subsets indicates that a basic two-signal program drives TRC/MRC/FDC subset specification and thereby shapes the distinct microenvironments in the white pulp. It appears that mLTo cell activation via the LTβR signaling functions as essential "signal 1" that needs to be delivered to initiate and to ensure specificity of the process. The existence of a mandatory first signal resembles the two-signal principle underlying lymphocyte activation, which requires the encounter with the cognate antigen as "signal 1"[52]. Accordingly, "signal 2" components need to be present to secure, for example, T-cell zone- vs. B-cell zone-specific reticular cell differentiation pathways. It is conceivable that second signals guiding reticular cell subset specialization reflect extrinsic imprints delivered by lymphocyte subsets in their immune microenvironments. Such context-specific integration of secondary differentiation signals by reticular cells may involve other members of the TNF receptor superfamily, which serve as secondary signals during T- and B-cell differentiation[53]. Interestingly, both naive T and B cells, i.e., the progenitors of effector lymphocytes, are maintained in the absence of specific antigenic stimulation by a process known as homeostatic proliferation[54,55]. Likewise, postnatal reticular progenitors were maintained and proliferated at a low frequency in the absence of LTβR signaling suggesting that yet undefined "signal 0" processes are sufficient to nourish the progenitor niches. Hence, while homeostatic signals are required to maintain the pool of reticular cell progenitor cells, the differentiation trajectories of reticular cell subsets reveal the fundamental two-signal principle that governs the structural and functional integrity of lymphoid organs.

The primary function of LTβR signaling in reticular cell differentiation has been demonstrated in mice that lack *Ltbr* expression in Ccl19-expressing cells[13], where reticular cell subsets fail to differentiate and remain arrested in the myofibroblastic progenitor stage. Second signals required for subset specification can be delivered by other molecules of the TNF receptor superfamily. For example, ablation of TNF in B cells prevents FDC differentiation[56], while the lack of CD30 contributes to abnormal B- and T-cell zone segregation that is associated with loss of podoplanin expression[15]. The massive extension of the complex receptor-ligand system of the TNF receptor superfamily that has accompanied the development of the adaptive immune system in jawed vertebrates[57] further supports the notion of an evolution of discrete signaling hierarchies in reticular cell differentiation. The advent of highly specialized immune environments in lymph nodes of mammals[48] has probably made LTβR the master regulator that controls the initiation of reticular cell subset specification. In conclusion, the concept that discrete signaling levels in perivascular niches determine the nature of reticular cell networks provides a theoretical framework for further exploration of

innate and adaptive immune processes in secondary and tertiary lymphoid tissues.

## Methods

**Mice**. All mouse strains were on a C57BL/6NCrl genetic background and maintained in individually ventilated cages under specific pathogen free conditions. The BAC-transgenic C57BL/6N-Tg(Ccl19-tTA)688BIAT (*Ccl19-tTA*) mouse was generated using homologous recombination-mediated transgenesis[58] using a tetracycline transactivator-based single-cassette system for spatiotemporal gene regulation[59]. The C57BL/6N-Tg(*Ccl19-Cre*)489Biat (*Ccl19-Cre*)[13] and the Ltbr^{tm1.1Thhe} (*LTβR^{fl/fl}*)[60] strains were described previously. B6.129 × 1-Gt(ROSA)26Sor^{tm1(EYFP)Cos}/J (R26R-EYFP) and B6.129 × 1-Gt(ROSA)26Sor^{tm1(CAG-Brainbow2.1)Cle} (Brainbow2.1)[32] mice were purchased from The Jackson Laboratories and the LC-1 strain[26] was kindly provided by Dr. Fendler, Max Delbrück Center of Molecular Medicine, Berlin, Germany. Fate-mapping experiments were performed via treating the pregnant dam with 1 mg doxycycline (Sigma Aldrich) intraperitoneally and maintenance of doxycycline in the drinking water (1 mg/ml). Experiments were performed in accordance with Swiss federal and cantonal guidelines (Tierschutzgesetz) under the permission numbers SG13/15, SG16/07 and SG05/17 granted by the Veterinary Office of the Canton of St. Gallen.

**Preparation of splenic stromal cells**. Spleens were harvested and perfused with RPMI 1640 medium containing 2% FCS, 20 mM Hepes (all from Lonza), 0.2 mg/ml Collagenase P (Sigma Aldrich), 0.8 U/ml Dispase I (Sigma Aldrich) and 100 µg/ml DNaseI (Applichem) with 22 G syringe. Samples were torn into smaller pieces and incubated at 37 °C for 30 min, with resuspension and collection of supernatant every 15 min to PBS containing 1% FCS and 10 mM EDTA (MACS buffer). To enrich fibroblastic stromal cells, hematopoietic and erythrocytes were depleted by incubating the cell suspension with MACS anti-CD45 and anti-TER119 microbeads (Miltenyi Biotec) and passing them through a MACS LS column (Miltenyi Biotec). Unbound single-cell suspensions were used for further flow cytometric analysis.

**Flow cytometry**. Single-cell suspensions were incubated for 20 min at 4 °C in PBS containing 1% FCS and 10 mM EDTA with the following antibodies: CD3 (clone:145-2C11, BioLegend at 1:200, Cat: 100305), CD19 (clone: eBio1D3, Thermo Fischer Scientific at 1:200, Cat: 25-0193-82), CD127 (clone:SB/199, BioLegend at 1:100, Cat: 121121), RORγt (clone:B2D, Thermo Fischer Scientific at 1:100, Cat: 46-6981-82), CD45 (clone:30-F11, BioLegend at 1:200, Cat: 103113), CD71 (clone: RI7217 at 1:200, Cat: 113811), CD157 (clone:BP-3, BioLegend at 1:200, Cat:140207), Ter119 (clone:TER-119, BioLegend at 1:200, Cat: 747740), CD31 (clone:MEC13.3, BioLegend at 1:200, Cat: 102419), PDPN (clone:8.1.1, BioLegend at 1:200, Cat: 127407 or 127417), CD35 (clone:8C12, BD Bioscience at 1:200, Cat: 553816), MAdCAM-1 (clone:MECA-367, BioLegend at 1:100, Cat: 120705), CD140a (clone:APA5, BioLegend at 1:100, Cat: 135909), Sca-1 (clone:D7, BioLegend at 1:100, Cat: 122523), CD29 (clone: HMbeta1-1, BioLegend at 1:200, Cat: 102225), CD106 (Clone: 429, BD Bioscience at 1:200, Cat: 740865), Endoglin (Clone: MJ7/18, BD Bioscience at 1:200, Cat: 564746), CD44 (Clone: IM7, BioLegend at 1:200, Cat: 103011). Streptavidin-conjugated PE-Cy7 (BioLegend at 1:1000, Cat: 405206) have been used to detect biotinylated antibodies. Cells were acquired with a FACS Canto II or BD LSR Fortessa (BD Biosciences) and analyzed using FlowJo software v7 or v10 (Tree Star Inc.).

**Histology**. Spleen samples were fixed overnight with 4% paraformaldehyde (Merck) in PBS under agitation at 4 °C. Fixed samples were washed with PBS containing 1% TritonX-100 (Sigma) and 2% normal serum (Sigma) overnight at 4 °C. Samples were embedded in 4% low-meting agarose gel (VWR International) and were sectioned into 40 µm using vibratome (Leica VT-1200). Sections were blocked in PBS containing 1% TritonX-100 (Sigma), 10% fetal calf serum (Sigma) and 1 mg/ml anti-Fcγ receptor (BD Biosciences) at 4 °C for 2 h and further incubated overnight with the following antibodies: anti-CD157 (BioLegend at 1:200, Cat:140207), anti-PDPN (BioLegend at 1:500, Cat: 127401), anti-CD4 (BioLegend at 1:500, Cat: 100403), anti-B220 (BioLegend at 1:500, Cat: 103229), anti-αSMA (Sigma at 1:500, Cat: C6198), anti-EYFP (Takara at 1:500, Cat:632592), anti-CCL21 (R&D Systems at 1:100, Cat:AF457), anti-CXCL13 (R&D Systems at 1:100, Cat: AF470), anti-MAdCAM-1 (clone:MECA-367, BioLegened at 1:100, Cat: 120705), anti-CD21/35 (BD Bioscience at 1:100), anti-CD35 (clone:8C12, BD Bioscience at 1:100, Cat: 553816), anti-TER119 (BioLegend at 1:500, Cat: 116203), anti-CD29 (clone: HMbeta1-1, BioLegend at 1:100, Cat: 102203), anti-Calponin-1 (clone: EP798Y, abcam at 1:100, Cat:ab46794), anti-Lumican (R&D systems at 1:200, Cat:AF2745), ant-Methoselin (Invitrogen at 1:100, Cat:PA5-79698), Endoglin (BD Bioscience at 1:100, Cat: 550546) and anti-tdTomato (Takara at 1:500, Cat:632496). Unconjugated antibodies were detected with the following secondary antibodies: Alexa488-conjugated anti-rabbit-IgG (at 1:500, Cat: 711-005-152), Dylight549-conjugated anti-rat-IgG (at 1:500, Cat: 712-165-153), Dylight549-conjugated anti-syrian hamster-IgG (at 1:500, Cat: 307-506-003), Cy3-conjugated Streptavidin (at 1:500, Cat: 016-160-084), Alexa647-conjugated Streptavidin (at 1:1000, Cat: 016-600-084), Alexa647-conjugated anti-goat-IgG (at 1:500, Cat: 705-606-147)

(all purchased from Jackson Immunotools). For Sca-1 and CD29 staining, splenic tissue was fixed for 2 h with 4% paraformaldehyde in PBS under agitation at 4°C. Samples were incubated in 30% sucrose overnight at 4 °C and were embedded in OCT (VWR International) and frozen on dry ice. OCT-embedded samples were sectioned into 6–8 µm thickness using a cryostat (MICROM HM 500 OM). Sections were blocked with PBS containing 1% TritonX-100 (Sigma), 10% fetal calf serum (Sigma) and 1 mg/ml anti-Fcγ receptor (BD Biosciences) at 4 °C for 2 h and further incubated overnight with biotin-conjugated anti-Sca-1 (BioLegend at 1:100, Cat: 108103) or Biotin-conjugated anti-CD29 (BioLegend at 1:100, Cat: 102203) followed by tyramide signal amplification kit (Life Technologies). For whole mount tissue acquisition, spleens were fixed overnight with 4% paraformaldehyde in PBS at 4 °C, washed twice with PBS containing 1% TritonX-100 (Sigma), 10% fetal calf serum (Sigma) and 1 mg/ml anti-Fcγ receptor (BD Biosciences) for 2 h and incubated with anti-αSMA-Cy3 antibodies overnight. After three more washing steps splenic tissues were cleared with FocusClear (CelExplorer) and directly imaged. Microscopy analysis was performed using a confocal microscope (Zeiss LSM-710) and images were processed with ZEN 2010 software (Carl Zeiss, Inc.) and Imaris (Bitplane).

**Quantification of Ki67+ cells by histology**. Spleens from Ccl19-iEYFP mice were sectioned at embryonic day E19.5, P0, P7, 2 week and 6 week. Samples were stained for the following markers: DAPI (Thermo Fischer Scientific), EYFP, α-SMA and Ki-67 (Clone; SolA15, Thermo Fischer Scientific at 1:500, Cat: 13-5698-82). One to five white pulp data sets were generated per mouse. 3D Z-stack images of the sections were acquired by high-resolution confocal microscopy. EYFP+ reticular cells were 3D reconstructed in Imaris (Bitplane) and the 3D isosurfaces were masked to the DAPI and Ki-67 channels separately in order to generate DAPI+ and Ki-67+ nuclei belonging to reticular cells specifically. Total numbers of DAPI+ reticular cells and Ki-67+ reticular cells per white pulp were calculated using the Spots object generation. Any false positives were manually removed from analysis. Likewise, false negatives were minimized by manually adding spots to the reticular cells that were not detected by the Spots algorithm in Imaris. The results of the analysis were displayed as percentage of activated Ki-67+ cells from the total number of reticular cells in the splenic white pulp.

**Splenic reticular cell sorting, library preparation and RNA-seq analysis**. EYFP+ cells from Ccl19-iEYFP fetal spleens (mean yield = 1150 cells, $n = 5$), adult Ccl19-iEYFP (mean yield = 4100 cells, $n = 7$) and Ccl19-iEYFP *Ltbr^{fl/fl}* (mean yield = 3500 cells, $n = 9$) spleens were sorted with a Bio-Rad S3 cell sorter and were collected in 200 µl Eppendorf tubes containing 100 µl of RNAlater reagent (Qiagen) to preserve the RNA after sorting. Likewise, reticular cell populations were sorted based on the defined markers sets, i.e., PDPN+ TRC (mean yield = 670 cells, $n = 4$), CD35+ FDC (mean yield = 950 cells, $n = 3$), MAdCAM-1+ MRC (mean yield = 1010 cells, $n = 2$), Sca-1+ PRC (mean yield = 1525 cells, $n = 3$). RNA extraction was performed using Quick-RNA Mini-Prep (Zymo Research). Reverse transcription and cDNA library generation was done using the Ovation SoLo RNA-Seq System (NuGEN) as described by the manufacturer. cDNA libraries were quantified by KAPA library Quantification kit (KAPA Biosystems) and Agilent Tape station in the Functional Genomics Center Zurich, Zurich, Switzerland. Sequencing of the cDNA libraries was performed on a Illumina HiSeq 2500 by the Functional Genomics Center Zurich.

**RNA-seq data processing and analysis**. Single-end reads (126 nt) were trimmed with Trimmomatic v0.33[61] and flexbar v2.5[62] to remove adapters and low-quality bases. The trimmed reads were aligned to the mouse genome (GRCm38) with STAR v2.5.2a[63] and deduplicated with NuDup v2.2. Reads that start at the same genomic coordinate and have the same strand orientation and 8nt molecular tag sequence were considered duplicates. The deduplicated BAM files were converted back to FASTQ format using bedtools bamtofastq v2.17.0 and duplicate read entries were removed. A transcriptome index was generated from the combined cDNA and ncRNA sequence files (Ensembl GRCm38.82) and Salmon v0.7.1[64] was used to estimate transcript abundance, which were read into R (v3.4.0) and summarized on the gene level with tximport (v1.4.0), together with average transcript length offsets[65]. Six samples with low-mapping rate from Salmon (<30%) and/or poor strand specificity were excluded from further analysis. For differential expression analysis, only genes with estimated CPM above 0.5 in at least 2 samples were retained and a generalized linear model with analysis date (batch) and cell type as predictors was fit to each gene using edgeR (v3.18.1)[66]. Two groups of contrasts were tested using edgeR's LRT framework: the first group contains the three pairwise comparisons between the embryonic EYFP+ cells, *Ltbr*-proficient and *Ltbr*-deficient EYFP+ adult cells, and the second group contains the ten pairwise comparisons between embryonic EYFP+ cells as well as the PRC, MRC, TRC, and FDC subsets. $p$ Values were adjusted for multiple comparisons using the Benjamini–Hochberg method within each contrast and genes with FDR-adjusted $p$ value < 0.05 were considered significantly differentially expressed. For each group of contrasts, MDS plots were generated from the logCPM values (calculated with the cpm function of edgeR with a prior count of 2) of the genes that were differentially expressed for at least one of the included contrasts (adjusted $p$ value < 0.05), using the plotMDS function of the limma package (v3.32.2). The camera

function from the limma package[67] was used to perform gene set analysis for the first group of contrasts, considering the C2, C5, and C7 gene-set collections from mSigDB (v5.2) downloaded from http://bioinf.wehi.edu.au/software/MSigDB/. The $p$ values were calculated using the competitive gene set test accounting for inter-gene correlation within the camera function. Individual gene expression heatmaps display logCPM values, incorporating TMM normalization factors, followed by gene-wise $Z$-score standardization and, where indicated, averaging over all samples within a group.

**Droplet-based single-cell RNA-seq analysis**. Total splenic fibroblastic stromal cells were sorted with a Bio-Rad S3 cell sorter based on the fibroblast markers ICAM-1, PDGFRβ, CD157, and PDPN from Ccl19-iEYFP adult spleens and single cell suspensions were run on the 10× Chromium analyzer (10× Genomics)[68]. Likewise, EYFP+ cells from Ccl19-iEYFP and Ccl19-iEYFP *Ltbr*^fl/fl adult spleens were sorted with a Bio-Rad S3 cell sorter and were run on the 10× Chromium analyzer (10× Genomics)[68]. The cDNA library generation was performed by the functional genomic center Zurich following the established commercial protocol for Chromium Single Cell 3′ Reagent Kit (v2 Chemistry). Libraries were run via Hiseq2500 rapid run for Illumina sequencing. Samples containing total splenic fibroblastic stromal cells and only EYFP+ cells were analyzed separately. Preprocessing and gene-expression estimation was performed using CellRanger (v2.1.1)[69], generating UMI counts for 8364 cells and 26,596 genes with a count above zero in at least one cell for samples containing total splenic fibroblastic stromal cells and 1852 cells and 24,796 genes from EYFP+ samples. The reference files used to build the index were obtained from the Ensembl GRCm38.90 release. Initial quality control was done with the scater R/Bioconductor package (v1.6.3)[70] running in R v3.4.2. In order to remove contaminating cells from total splenic FRC samples, cells expressing one of the markers *Lyve1, Hba-a1, Hba-a2, Ptprc, Tfrc, and Cldn5* were excluded from further analysis, reducing the cell number to 6856 cells. Similarly, in EYFP+ samples, 276 cells expressing at least one of the markers *Lyve1, Hba-a1, Hba-a2,* or *Krt18* were considered as non-FRC and excluded from further analysis. In addition, cells with exceedingly high or low number of detected genes or UMI counts (more than 2.5 median absolute deviations from the overall median, in either direction) were excluded, as were all cells with a large fraction of mitochondrial reads (more than 2.5 median absolute deviations above the median fraction). After these QC steps, 6227 cells were retained in total splenic FRC samples and 1318 cells were retained for EYFP+ cell samples. Genes that were not assigned at least 1 read in at least two of these cells were filtered out, leaving 22,619 genes in total splenic FRC samples and 20,146 genes in EYFP+ samples. Next, we used Seurat package in R (v2.3.0)[71] to normalize the UMI counts, regress out the influence of the number of UMI counts per cell, and find highly variable genes. We also performed dimension reduction with PCA and t-SNE as well as cell clustering using Seurat for all datasets. For the EYFP+ samples, all analyses were performed both for the full dataset and for the subsets with *Ltbr*-proficient and *Ltbr*-deficient EYFP+ cells from adult spleens. In each of the datasets, one of the obtained clusters was inferred to contain both TRC and MRC cells and was further clustered into two subgroups using Seurat. Using canonical FRC markers and gene signatures from pop-RNA-seq, each cluster was assigned an informative label. The normalized expression values from Seurat were used to visualize expression levels of genes of interest across the inferred clusters as well as to find marker genes for each cluster, using the Wilcoxon test implemented in the Seurat package. In addition, for each cluster in the EYFP+ samples, we used these values to calculate a signal-to-noise ratio (SNR) statistic for each gene, comparing the cells in the cluster to all cells outside the cluster, and thus generate a ranked list of genes for each cluster. These ranked lists were then supplied to GSEA-Preranked (v3.0) to investigate the enrichment of gene sets from the mSigDB (v5.2) collections C2, C5, and C7 downloaded from http://bioinf.wehi.edu.au/software/MSigDB/.

**Differentiation trajectory analysis**. In order to resolve differentiation trajectories we used the Monocle2 (v2.6.4) (PMID 28825705) package in R with normalized expression values from Seurat as input. The combination of the top 200 marker genes for each cluster (based on their logFC values) were selected for unsupervised ordering of the cells and the DDRTree algorithm implemented in Monocle2 was applied for trajectory reconstruction. To further resolve the differentiation towards TRC/MRC/FDC clusters, we reran the analysis on all cells that were aligned along the LTβR-dependent branch starting from the branching point.

**Statistics**. GraphPad Prism 7 was used for all statistical analyses. Differences with a $p$ value < 0.05 were considered statistically significant.

**Reporting summary**. Further information on experimental design is available in the Nature Research Reporting Summary linked to this article.

## Data availability

RNAseq and scRNA-seq data are deposited in the arrayexpress database (www.ebi.ac.uk/arrayexpress) under accession numbers E-MTAB-7703, E-MTAB-7097, and E-MTAB-7094. All other data are available from the authors upon reasonable requests.

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

## Acknowledgements

This study received financial support from the Swiss National Science Foundation (Grants 166500 and 159188 to B.L.), the Human Frontiers Science Program (RGP0034/206), and the Deutsche Forschungsgemeinschaft (HE3116/9-1 to T.H. and B.L.). The funders had no role in study design, data collection and analysis, decision to publish, or preparation of the paper.

## Author contributions

B.L. designed the study, discussed the data and wrote the paper; H.-W.C. performed the experiments, analyzed the data and wrote the paper; L.O., M.N., C.S., N.P., M.L. and E.S. performed the experiments, analyzed and discussed the data; J.M., A.T., U.S. and K.P. provided the reagents; T.R. produced the transgenic mice and discussed the data, M.R. and T.H. supported the study design and discussed the data.

## Additional information

**Competing interests:** The authors declare no competing interests.

