## [Peer Review File · Nature Communications]

Reviewers' comments:

Reviewer #1 (Lymphoid organogenesis, LTb signalling)(Remarks to the Author):

In this study, a novel Ccl19-iEYFP mouse model was created to examine the origin and differentiation splenic reticular cells. A major finding is that all reticular cell subsets were derived from mLTo subset at periarterial site of E19.5 stage. The concept that discrete signaling levels in perivascular niches determine the nature of reticular cell networks provides an interesting model for further exploration of innate and adaptive immune processes in secondary and tertiary lymphoid tissues. The ms will be further improved by addressing following issues

1. Given the probably biased labeling using Ccl19-promoter, one concern is that at which degree this is true for the development of all the reticular cells in the spleen.
2. How much percentages of adult reticular cell subsets are derived from the mLTo progenitors if perinatal labeling is performed? If this mLTo subset is ablated using DTR system, does this abolish all subsequent development of related reticular cell subsets?
3. LTbR-independent and -dependent stages were proposed based on the current results, however, only on percentage data. Are the absolute numbers of mLTo, PRC changed in the absence of LTbR? This need to be done on the global LTbR knockout background, given the undetermined deletion efficiency of LTbR on Ccl19-transgenic mice. Actually, the deletion efficiency should be determined since mLTo stage.
4. A technical concern: Cell labeling is currently performed using a tet-off system, which heavily relies on the tet-off efficiency. Given Ccl19 is also constantly expressed on certain type of cells, without stringent tet-off efficiency, it is difficult to tell the labeled cells from the constantly-expressing cells. A tet-on system can more reliably label cells at a specific time-frame to determine their progenies.

Reviewer #2 (Endothelial differentiation, fate mapping, immune imaging)(Remarks to the Author):

In this study, Cheng et al used a novel mouse model to trace CCL19 expressing cells and their progeny at different developmental time points. They suggest that a perivascular subset of cells called « PRC » acts as the progenitor of all splenic mesenchymal cell types in a Ltbr dependent fashion. The manuscript is well written and the model appears cleverly designed. The major conclusion is supported by solid data, although I have some concerns.

Major concerns :

- 1/ The number of fate-mapped lineages is not clear to me. It appears that all the capsular cells (adventitial ? smooth muscle ? fibroblasts ? mesothelium?) are labeled, as well as the perivascular cells of the red pulp. The number and nature of fate-mapped lineages need further clarification (both by flow and most importantly imaging).
- 2/ I am confused by the fact that PRC would be more numerous than TRC, FDC or MRC (flow data shown in Fig 1h and 3i). It is clear from the imaging data presented in this manuscript as well as from other studies that TRC and FDCs outnumber PRC. How the authors reconcile this? Could it be that PRC represent a mixture of cells, including the ones in the capsule and the perivascular cells of the Red Pulp (plus other cell type(s)?)
- 3/ Lineage tracing experiments would greatly benefit from multicolor lineage tracing (confetti-like) models to prove the lineage relationship of the subsets. This would in addition exclude the possibility that FRC, FDC or MRC emerge from the capsule or the red pulp perivascular cells.

4/ Are « PRC » the progenitors of splenic stromal cells subsets during an immune response, when the TRC network is compromised (LCMV response e.g)? The authors have published several articles using this model, this should thus be easily done and would greatly strengthen the study.

5/ The idea of a perivascular cell type acting as a progenitor for other stromal cells types in the spleen is not new. Krautler et al (Cell. 2012 Jul 6;150(1):194-206) and more recently Schaeuble et al (Cell Rep. 2017 Nov 28;21(9):2500-2514) have suggested that. Schaeuble et al -and this study- both agree on the Ltb-dependent differentiation of TRC from perivascular reticular cells. However, I'm puzzled by the fact that Schaeuble et al. demonstrated that CXCL13 expressing cells can still be found in Ltb-deficient P5.5 mice around the central arterioles, whereas the authors show here that FDCs differentiation from PRCs is LtbR dependent (figure 3g and 6b-c). Furthermore, the authors published that CXCL13 expression was not affected in LN of adult Ccl19-cre LtbRfl/fl mice (Immunity. 2013 May 23;38(5):1013-24). Can the authors discuss the nature of this discrepancy? To resolve this issue, the authors could analyze the presence of FDCs, CXCL13 expressing cells and B cells by imaging in E0-E19.5 LtbRfl/fl fate-mapped mice at P5.5. In any case, the Schaeuble et al article MUST be acknowledged.

6/ The authors indicate that YFP+ cells constitute only a minor fraction (0,31%) of nonendothelial cells (supplementary figure 1e). However, this does not look like to be the case by imaging. In fig 4e, all WP mesenchymal cells seem to be fate mapped and it's hard to imagine that they represent only 0.31% of all splenic, non endothelial stromal cells (Fig 4e). Option 1: many WP stromal cells are indeed YFP-cells. If this is the case, another precursor can massively differentiate into WP stromal cells and the take-home message of the study is incorrect. In order to test this, the authors should show the % of YFP+ cells in each fraction of WP stromal cells (FRC, FDC, MRC, etc...) instead of gating first on YFP+ cells and asking the cellular composition of this group. Immunostainings could also help identifying/quantifying YFP- stromal cells in each population. Option 2: if all WP, non endothelial stromal cells are indeed fate mapped, it means that the so-called "DN" CD45- CD31- fraction is not stromal but contaminated by hematopoietic cells, just as recently shown in the BM by Frenette (Immunity. 2018 Oct 16;49(4):627-639). In the end, flow data and imaging data should fit. NB: This low % of lineage tracing (0.31%) is observed in mice that have not been treated with Dox, ie in a situation where the labeling is maximal. I am thus concerned by Fig 3.i. Since this strategy labels a tiny fraction of stromal cells (probably 1 or 2% of all WP stromal cells based on a rough estimation, ie 1 or 2% of 0.31%...), how many cells were analyzed to create this panel?

7/ In the same line, in figure S1d, the presence of a tdTomato+ YFP- population in the « -Dox panel » suggests that some cells did not recombine the EYFP allele efficiently, but do express the tdTomato reporter. It should be determined by imaging if a tdTomato+ EYFP- population can be found in the white pulp or the red pulp of adult mice. Of note, the line between tdTomato+ and tdTomato- cells in this panel seems to be randomly drawn to get only tdTomato+ YFP+ cells in non Dox treated mice. A control mouse bearing all the transgenes of the CCL19-iEYFP but not the CCL19i-tTA tg is mandatory. All these technical questions are key because they will greatly help to define this new model and avoid the issues observed in CCL19Cre mice originally described to be « FRC specific »...

Minor points :

- Ter119 labeling prevents the reader to assess YFP expression in the red pulp or the capsule. A supplemental figure dedicated to this purpose would help.
- The fact that the PRC giving rise to TRC/FDC and MRC express Sca1 is not shown and cannot be claimed in the abstract. Only a Sca1 staining in fig 1f, 2a and 3 would convince me that the few GFP

cells labeled at E19.5 are expressing Sca1.

- In figure 1, the legend should indicate that these mice were never treated with dox.
- In figure 1g, the YFP+ pdpn+ CD35- population is referred to as TRC, but MRCs also express pdpn. Using the tdTomato reporter would confirm the TRC gating strategy.
- The conclusion from figure 1h (and 3i) is that LtbR-KO fate-mapped cells accumulate as PRCs, suggesting that these PRCs are precursors of WP stromal cells. However, the data are represented as percent of YFP+ cells, and cell counts are needed to confirm the "accumulation" of PRCs in LTbR-KO fate-mapped cells. This would strengthen the conclusion that PRCs are WP stromal cells precursors, and that differentiation of TNCs is LTbR-independent.
- In figure 3, E0-E19.5 fate mapped mice analyzed at adult age reveals that PRCs contribute to adult TRCs, MRCs and FDCs. TRCs are only defined by pdpn expression whereas determining the functional state of these cells could be easily determined by analyzing tdTomato in these cells. This would confirm that E0-E19.5 fate-mapped cells contribute to functional TRCs. Expression of CXCL13 could also be used in MRCs and FDCs.
- In figure S4f, the authors show the imprinted gene network of YFP+ cells at different stages and in different conditions. But there are few hundreds of imprinted genes in the mouse. Why/how were these ones selected? What is the behavior of the other ones? If it is a general rule, it might be important to confirm this more thoroughly in a dedicated sup figure (or a bigger panel). If not, please remove the panel.

Point by point reply

We would like to thank the Editor and Reviewers for their thoughtful comments on our manuscript. We very much appreciated that reviewers found merit in our work and have judged the results of our study as an “*interesting model for further exploration of innate and adaptive immune processes in secondary and tertiary lymphoid tissues*” (R1) and that the novel model system is seen as “*cleverly designed*” (R2). We have carefully addressed all concerns of the reviewers with novel data including single cell RNA sequencing data from splenic fibroblasts (red pulp, white pulp and capsule) and have amended our manuscript accordingly. Our responses to the reviewers’ comments have been outlined in the following point-by-point reply.

Reviewer 1:

1. *Given the probably biased labelling using Ccl19-promoter, one concern is that at which degree this is true for the development of all the reticular cells in the spleen.*

We agree that the *Ccl19* promoter targets a distinct fraction of splenic fibroblastic stromal cells. The functional analysis based on cell type-specific genetic perturbation revealed that the *Ccl19*-iEYFP transgenic model facilitates targeting of those fibroblastic stromal cell populations that build and maintain the splenic white pulp infrastructure. To address the concern and to provide a broader view on the splenic fibroblastic stromal cell landscape, we performed a single-cell RNA sequencing (scRNA-seq) analysis. The results are displayed in the new Figure 5a-d and Supplementary Figure 6 and are described in the respective section of the revised manuscript on page 10. In brief, expression of EYFP mRNA is almost exclusively restricted to white pulp reticular cells, while only sporadic transgene activity in red pulp fibroblasts, subcapsular fibroblasts and mesothelial cells could be detected. Importantly, we found that Endoglin marks both red pulp fibroblasts and PRCs (new Fig. 5c and Supplementary Fig 5a). Taken together, this additional unbiased scRNA-seq analysis confirms that the *Ccl19*-iEYFP model is well-suited for the specific targeting of white pulp reticular cells and hence permits the delineation of the differentiation trajectories of these cells. See also our detailed comments to points 1 and 2 raised by Reviewer 2.

2. *How much percentages of adult reticular cell subsets are derived from the mLTo progenitors if perinatal labeling is performed? If this mLTo subset is ablated using DTR system, does this abolish all subsequent development of related reticular cell subsets?*

We performed the suggested cell fate-mapping experiment in *Ccl19*-iEYFP mice starting from postnatal day 2 (P2) to 6 weeks of age. Confocal microscopic analyses revealed that a fraction of differentiated FRC subsets were labeled with the EYFP transgene (Fig. R1a and b for the attention of the reviewer). Based on the alternative flow cytometric approach that excludes CD71-positive erythrocyte precursor cells and uses gating on ICAM-1⁺ CD45⁻ Ter119⁻ cells (Fig. R1c and d, see also comment 6 of Reviewer 2 and Supplementary Fig. 5d), about 17% of splenic fibroblastic stromal cells are labelled by the transgene in the absence of doxycycline treatment, while approximately 4% of all fibroblastic stromal cells are targeted in the P2 to 6 week fate-mapped condition (Fig. R1d). This analysis confirmed that P2 transgene-positive can give rise to all white pulp reticular cell subsets including Endoglin-positive PRCs (Fig. R1c), hence corroborating the data shown in the main Figures 1 and 3. Thus, mLTo cell commitment starts at E19.5 and continues along the growing arteriolar tree during the first week of postnatal development.

The ablation of reticular cells using the diphtheria toxin system (*Ccl19*-Cre iDTR) lasts maximally 3-4 weeks in adult mice (Novkovic et al. 2016, PMID 27415420). Since proliferation and most likely also turnover of reticular cells in young mice is accelerated (see Fig. 2f), DT-mediated

ablation of neonatal mLT α cells will most likely only be effective for a few days, hence impeding the assessment of any progeny-progenitor relationship of perinatal mLT α cells and adult reticular cell subsets. Nevertheless, the cell fate-mapping experiment in *Ccl19-iEYFP Ltb^r^{fl/fl}* mice from postnatal P2 to adult revealed that there is no significant difference in the percentage of transgene-positive cells under *Ltb^r*-proficient or -deficient conditions (R1c and d). In addition, the white pulp reticular cell specification was abrogated in the absence of LT β R-signaling (Fig. R1e), corroborating the data shown in the main Figure 3. The results of the P2-to-adult fate mapping further supports the conclusion that the maintenance of PRCs is independent of LT β R-signaling.

Figure R1. Fate-mapping of progenitor cells from postnatal day 2 to 6 weeks in *Ccl19-EYFP* and *Ccl19-iEYFP Ltb^r^{fl/fl}* spleens. (a-b) Confocal microscopy analysis of spleens harvested from *Ccl19-iEYFP* mice treated with doxycycline from postnatal day 2 onwards to 6 weeks old (scale bars in d = 500 μ m, scale bar in e = 50 μ m). Boxed areas in (a) indicate regions of higher magnification shown in right-hand panels with scale bars = 10 μ m. (c) Representative flow cytometric analysis of fibroblastic stromal cells from the indicated mouse strains without doxycycline treatment or with doxycycline treatment from P2 to 6 weeks. (d) Percentage of EYFP⁺ cells of ICAM-1⁺ fibroblastic stromal cells in spleens of the indicated mouse strains and fate-mapping conditions based on gating in (c). (e) Percentage of ICAM-1⁺EYFP⁺ cells from spleens based on gating in (c) (n = 5-8 mice per group from 3 independent experiments, mean \pm SEM).

3. *Ltbr*-independent and -dependent stages were proposed based on the current results, however, only on percentage data. Are the absolute numbers of mLT_o, PRC changed in the absence of *Ltbr*? This need to be done on the global *Ltbr* knockout background, given the undetermined deletion efficiency of *Ltbr* on *Ccl19*-transgenic mice. Actually, the deletion efficiency should be determined since mLT_o stage.

We have recorded the PRC numbers in globally *Ltbr*-deficient mice in comparison with wildtype control and *Ccl19*-iEYFP *Ltbr*^{fl/fl} mice. As expected, there was no significant difference in PRC cell numbers between *Ltbr*-deficient and *Ccl19*-iEYFP *Ltbr*^{fl/fl} mice (Fig. R2). These results indicate that the *Ccl19*-iEYFP *Ltbr*^{fl/fl} mouse model recapitulates the phenotype of *Ltbr*-deficient mice in the spleen.

The efficacy of Cre recombinase activity is monitored in the *Ccl19*-iEYFP model through excision of the transcriptional stop cassette in the R26R-EYFP strain at all times, i.e. from the mLT_o cell stage to differentiated FRC subsets in the adult.

Figure R2. PRC numbers in *Ltbr*-deficient and *Ccl19*-iEYFP *Ltbr*^{fl/fl} mice. (a) Representative flow cytometric analysis of PRCs from the indicated mouse strains without doxycycline treatment and analyzed for the indicated markers. (b) Cell numbers of PRCs in *Ltbr* global knockout, *Ccl19*-iEYFP *Ltbr*^{fl/fl} mice and littermate controls ($n=5-7$ per group from 3 independent experiments).

4. A technical concern: Cell labeling is currently performed using a tet-off system, which heavily relies on the tet-off efficiency. Given *Ccl19* is also constantly expressed on certain type of cells, without stringent tet-off efficiency, it is difficult to tell the labeled cells from the constantly-expressing cells. A tet-on system can more reliably label cells at a specific time-frame to determine their progenies.

We have assessed the dose requirements and application regimens of doxycycline to permit constant and efficient inhibition of the Cre recombinase activity. Essential controls have been documented in the manuscript in the following display items:

- Supplementary Figure 1e shows that lifelong application of doxycycline completely abrogates the Cre-mediated recombination, but does not affect the *Ccl19* promoter activity as revealed by the absence of EYFP-positive cells in the flow cytometric analysis and the tdTomato expression mainly in T cell zone reticular cells.
- Supplementary Figure 3i: The application of doxycycline at embryonic day E17.5, i.e. before the *Ccl19* promoter shows activity in the embryonic spleen, revealed a complete absence of EYFP-positive cells in the adult spleen.

We have carefully considered the choice of the inducible gene expression system and show here that the tet-off system facilitates stringent control of gene expression in FRC progenitors. Hence, we feel that assessment of an alternative approach such as the tet-on system would be beyond the scope of this study.

Reviewer 2:

1. *The number of fate-mapped lineages is not clear to me. It appears that all the capsular cells (adventitial ? smooth muscle ? fibroblasts ? mesothelium?) are labeled, as well as the perivascular cells of the red pulp. The number and nature of fate-mapped lineages need further clarification (both by flow and most importantly imaging).*

To address this concern, we have isolated and analyzed fibroblastic stromal cells from Ccl19-iEYFP spleens by scRNA-seq. This approach facilitated the assessment of EYFP transgene activity in different splenic stromal cell compartments. The results of this experiment are now displayed in the revised Figure 5a-d and Supplementary Figure 5. Based on their specific gene signatures, we were able to identify three additional splenic fibroblastic stromal cell populations: red pulp fibroblasts, subcapsular fibroblasts and mesothelial cells. Importantly, the analysis confirmed that the Ccl19-iEYFP transgene targets mainly white pulp FRC. This additional scRNA-seq analysis in combination with validation by confocal microscopy and flow cytometry provides a comprehensive analysis of the fibroblastic stromal cell landscape in the murine spleen.

2. *I am confused by the fact that PRC would be more numerous than TRC, FDC or MRC (flow data shown in Fig 1h and 3i). It is clear from the imaging data presented in this manuscript as well as from other studies that TRC and FDCs outnumber PRC. How the authors reconcile this? Could it be that PRC represent a mixture of cells, including the ones in the capsule and the perivascular cells of the Red Pulp (plus other cell type(s)?)*

The high abundance of PRC in the splenic white pulp is most likely due to the heavy vascularization of this lymphoid organ that receives 5-10% of the cardiac output. The arterial blood pours into the marginal sinuses that surround the white pulp. This area is populated by Endoglin-expressing EYFP⁺ PRC (new Supplementary Fig. 5a). The new scRNA-seq analysis of splenic fibroblastic stromal cells (Fig. 5b-d) confirmed that Endoglin mRNA (*Eng*) expression is confined to red pulp fibroblast cluster 7 and overlaps with the expression of the perivascular myofibroblast markers *Pdgfrb* and *Ly6a* (Fig. 5c). Importantly, a small fraction of EYFP mRNA expressing cells appears in Cluster 7 suggesting that EYFP⁺ PRC are closely related to EYFP⁻ red pulp fibroblasts. Clearly, EYFP⁺ PRC are located around blood vessels within the white pulp and in the highly vascularized marginal sinus region. Cells in the splenic capsule appear to be either Lumican⁺ subcapsular fibroblasts appearing in clusters 10 and 11 or as Mesothelin⁺ mesothelial cells in clusters 8 and 9 (Fig. 5b-d and Supplementary Fig. 5c). These clusters harbor very few EYFP mRNA-expressing cells (Fig. 5d). Moreover, E19.5-to-6wk fate mapping revealed that these cells do not originate from early embryonic mLT₀ cells (Fig. R3).

3. *Lineage tracing experiments would greatly benefit from multicolor lineage tracing (confetti-like) models to prove the lineage relationship of the subsets. This would in addition exclude the possibility that FRC, FDC or MRC emerge from the capsule or the red pulp perivascular cells.*

To address this concern, we followed a two-pronged approach. First, we examined the transgene activity at E19.5 using co-staining with anti-Mesothelin. As expected, anti-Mesothelin antibodies labelled capsular cells in the fetal (Fig. R3a) and in the adult spleen (Supplementary Fig. 6c). Importantly, an overlap of EYFP and Mesothelin expression could not be detected in the E19.5 to adult fate-mapped spleen (Fig. R3b) indicating that the mesothelial capsular cells do not arise from the embryonic mLT₀ cells.

In the second approach, we have used the Brainbow2.1 model to determine whether capsular fibroblastic stromal cells originate from the embryonic mLT₀ cells. As shown in the new Supplementary Figure 3d, the differentiated white pulp FRC subsets can be labeled using the multi-color fate-mapping system. Importantly, E19.5 to 6 weeks fate-mapping confirmed that neither mesothelial cells nor subcapsular fibroblasts originate from the embryonic mLT₀ cells.

Figure R3. Identification of capsular cells in the spleen at different developmental stages. Spleens were harvested from *Ccl19-iEYFP* embryos at E19.5 (a) or from *Ccl19-iEYFP* mice treated with doxycycline from E19.5 onwards to 6 weeks old (b). Sections were stained with the indicated antibodies and analyzed by confocal microscopy (scale bars in a = 300 μm , scale bar in b = 500 μm). Boxed areas in (a) indicated region of higher magnification shown in right-hand images (scale bars = 100 μm). (n = 3 in each conditions)

4. Are « PRC » the progenitors of splenic stromal cells subsets during an immune response, when the TRC network is compromised (LCMV response e.g)? The authors have published several articles using this model, this should thus be easily done and would greatly strengthen the study.

We thank the reviewer for this interesting comment. While such experiments would be possible, this exhaustive analysis would exceed the scope of the current manuscript with the clear focus on the embryonic origin of splenic white pulp FRC. In fact, we are currently studying the progenitor-progeny relationship of adult PRC with the differentiated FRC subsets in lymph nodes. We feel that elaborating the differentiation trajectories of reticular cells under conditions of inflammatory perturbation deserves to be addressed in full detail in an independent body of work.

5. The idea of a perivascular cell type acting as a progenitor for other stromal cells types in the spleen is not new. Krautler et al (*Cell*. 2012 Jul 6;150(1):194-206) and more recently Schaeuble et al (*Cell Rep*. 2017 Nov 28;21(9):2500-2514) have suggested that. Schaeuble et al -and this study- both agree on the *Ltb*-dependent differentiation of TRC from perivascular reticular cells. However, I'm puzzled by the fact that Schaeuble et al. demonstrated that CXCL13 expressing cells can still be found in *Ltb*-deficient P5.5 mice around the central arterioles, whereas the authors show here that FDCs differentiation from PRCs is *LtbR* dependent (figure 3g and 6b-c). Furthermore, the authors published that CXCL13 expression was not affected in LN of adult *Ccl19-cre LtbRfl/fl* mice (*Immunity*. 2013 May 23;38(5):1013-24). Can the authors discuss the nature of this discrepancy? To resolve this issue, the authors could analyze the presence of FDCs, CXCL13 expressing cells and B cells by imaging in E0-E19.5 *LtbRfl/fl* fate-mapped mice at P5.5. In any case, the Schaeuble et al article MUST be acknowledged.

Indeed, not only FDC, but also other CD21/35-negative reticular cells show CXCL13 expression on the protein level (Supplementary Fig. 1c). Population-based RNA-seq analysis (Supplementary Fig. 4) and scRNA-seq analysis (Supplementary Fig. 6b) confirmed that both FDC and MRC produce *Cxcl13* mRNA. Interestingly, scRNA-seq analysis from *Ccl19-iEYFP LtbR^{fl/fl}* mice revealed that the intermediate PRC subset showed low level *Cxcl13* expression (Supplementary

Fig. 6b) indicating that *Ltbr*-deficiency reduces *Cxcl13* promoter activity, but does not completely abrogate the activity of this gene. This could explain some residual CXCL13 staining in the work of Schaeuble et al. As suggested by the reviewer, we followed up on this and have analyzed spleens harvested from E19.5-P5.5 fate-mapped *Ccl19-iEYFP Ltbr^{fl/fl}* and in controls spleens. We can confirm that lower CXCL13 protein expression can be detected in *Ltbr*-deficient spleens compared to the *Ltbr*-proficient situation (Fig. R4a). Although CXCL13 expression could be detected at P5.5, the signature FDC markers CD21/35 were not found at this time point (Fig. R4b). These results partially explain the previous finding that *Cxcl13* mRNA expression in lymph nodes of *Ccl19-Cre Ltbr^{fl/fl}* mice was not completely lost, but only down-regulated. Moreover, the *Ccl19-Cre* transgene spares a fraction of B cell zone FRC. We are currently investigating the properties of *Cxcl13*-expressing lymph node FRC in detail using the *Cxcl13-Cre/tdTomato* transgenic model (Onder et al. 2017, PMID 28709801). We anticipate to address and to discuss the phenotypical and developmental relationship of *Cxcl13*-expressing FRC in detail in a separate study.

We would like to emphasize that the Schaeuble 2017 and the Krautler 2012 publications have been cited in our manuscript as references 28 and 39, respectively.

Figure R4. Identification of FDC and CXCL13 expression in E19.5 to P5.5 fate-mapped spleens. (a-b) Spleens were harvested from *Ccl19-iEYFP Ltbr^{fl/fl}* mice treated with doxycycline from E19.5 onwards to P5.5. Sections were stained with indicated the antibodies and analyzed by confocal microscopy (Scale bars = 30 μ m).

- The authors indicate that YFP+ cells constitute only a minor fraction (0,31%) of nonendothelial cells (supplementary figure 1e). However, this does not look like to be the case by imaging. In fig 4e, all WP mesenchymal cells seem to be fate mapped and it's hard to imagine that they represent only 0.31% of all splenic, non endothelial stromal cells (Fig 4e). Option 1: many WP stromal cells are indeed YFP- cells. If this is the case, another precursor can massively differentiate into WP stromal cells and the take-home message of the study is incorrect. In order to test this, the authors should show the % of YFP+ cells in each fraction of WP stromal cells (FRC, FDC, MRC, etc...) instead of gating first on YFP+ cells and asking the cellular composition of this group. Immunostainings could also help identifying/quantifying YFP- stromal cells in each population. Option 2: if all WP, non endothelial stromal cells are indeed fate mapped, it means that the so-called "DN" CD45- CD31-fraction is not stromal but contaminated by hematopoietic cells, just as recently shown in the BM by Frenette (Immunity. 2018 Oct 16;49(4):627-639). In the end, flow data and imaging data should fit.

NB: This low % of lineage tracing (0.31%) is observed in mice that have not been treated with Dox, ie in a situation where the labeling is maximal. I am thus concerned by Fig 3.i. Since this strategy labels a tiny fraction of stromal cells (probably 1 or 2% of all WP stromal cells based on a rough estimation, ie 1 or 2% of 0.31%...), how many cells were analyzed to create this panel?

We would like to thank the reviewer for this important comment. To address this concern, we have established an alternative flow cytometry protocol using exclusion of CD71⁺ erythrocyte progenitors combined with gating on ICAM-1⁺ cells (Supplementary Fig. 5d and Fig. R1). The scRNA-seq analysis of splenic fibroblastic stromal cells revealed that Endoglin is a well-suited marker for red pulp fibroblasts and has therefore been incorporated in the flow cytometric analysis. This extended analysis revealed that more than 80% of the differentiated white pulp fibroblasts express the EYFP marker (Supplementary Fig. 5e). The frequency of EYFP⁺ cells within the ICAM-1⁺ splenic fibroblasts is around 17% (Fig. R1).

- 7. In the same line, in figure S1d, the presence of a tdTomato⁺ YFP⁻ population in the « -Dox panel » suggests that some cells did not recombine the EYFP allele efficiently, but do express the tdTomato reporter. It should be determined by imaging if a tdTomato⁺ EYFP⁻ population can be found in the white pulp or the red pulp of adult mice. Of note, the line between tdTomato⁺ and tdTomato⁻ cells in this panel seems to be randomly drawn to get only tdTomato⁺ YFP⁺ cells in non Dox treated mice. A control mouse bearing all the transgenes of the CCL19-iEYFP but not the CCL19i-tTA tg is mandatory. All these technical questions are key because they will greatly help to define this new model and avoid the issues observed in CCL19Cre mice originally described to be « FRC specific »...*

To address this question, we have performed further high resolution confocal microscopic analyses to locate EYFP⁻ and tdTomato-expressing cells. EYFP⁺ cells can be mainly detected in all splenic white pulp fibroblastic stromal cells, while the tdTomato expression was confined mainly to the T cell zone and fewer cells in the marginal zone (Fig. R5a and Supplementary Fig. 1e). Using the amended flow cytometry protocol, we found that around 50% of the EYFP⁺ cells co-express tdTomato, which corresponds to roughly 10% of ICAM-1⁺ fibroblasts (Fig. R5b). Importantly, less than 2% of ICAM-1⁺ cells express tdTomato alone (Fig. R5b). It is likely that tdTomato single-positive reticular cells have not been able to generate sufficient tTA protein to drive Cre-mediated recombination or that recombination has not yet taken place. Overall, the combination of our flow cytometric and confocal microscopic analyses combined with the scRNA-seq data clearly indicate that the Ccl19-iEYFP model provides a high specificity for differentiated splenic white pulp FRC and white pulp PRC.

Figure R5. Assessment of tdTomato expression in *Ccl19-iEYFP* spleens. (a) Confocal microscopic analysis of splenic cross sections stained with the indicated markers (scale bar = 300 μ m). Boxed areas in (a) indicate the region of higher magnification shown in right-hand images (scale bars = 200 μ m). (b) Representative flow cytometric analysis of transgene-positive cells analyzed with the indicated markers ($n > 3$ mice per group from 2 independent experiments). (c-d) Confocal microscopic analysis of splenic cross sections stained with the indicated markers (scale bar = 500 μ m). Boxed areas in (c and d) indicate regions of higher magnification shown in lower images (scale bars = 5 μ m; $n > 3$ mice).

Minor points :

1. *Ter119* labeling prevents the reader to assess YFP expression in the red pulp or the capsule. A supplemental figure dedicated to this purpose would help.

Supplementary Fig.1d has been amended accordingly.

2. The fact that the PRC giving rise to TRC/FDC and MRC express *Sca1* is not shown and cannot be claimed in the abstract. Only a *Sca1* staining in fig 1f, 2a and 3 would convince me that the few GFP cells labeled at E19.5 are expressing *Sca1*.

The abstract has been modified.

3. In figure 1, the legend should indicate that these mice were never treated with dox.

Done as suggested.

4. In figure 1g, the YFP+ pdpn+ CD35- population is referred to as TRC, but MRCs also express pdpn. Using the tdTomato reporter would confirm the TRC gating strategy.

Based on the confocal microscopy analysis, PDPN is not a specific marker for the TRC population but also labels mesothelial cells in the capsular region in the spleen. However, unlike lymph nodes, MRCs in the splenic white pulp do not express PDPN. Instead, tdTomato expression partially labelled the MRC population based on the confocal microscopic analysis (Fig. R5c and d). Since the EYFP transgene does not target mesothelial cells, we prefer to continue to use PDPN⁺CD35⁻ gating for EYFP⁺ TRC.

5. The conclusion from figure 1h (and 3i) is that LtbR-KO fate-mapped cells accumulate as PRCs, suggesting that these PRCs are precursors of WP stromal cells. However, the data are represented as percent of YFP+ cells, and cell counts are needed to confirm the "accumulation" of PRCs in LTbR-KO fate-mapped cells. This would strengthen the conclusion that PRCs are WP stromal cells precursors, and that differentiation of TNCs is LTbR-independent.

Absolute numbers are now shown in Supplementary Fig. 3c.

6. In figure 3, E0-E19.5 fate mapped mice analyzed at adult age reveals that PRCs contribute to adult TRCs, MRCs and FDCs. TRCs are only defined by pdpn expression whereas determining the functional state of these cells could be easily determined by analyzing tdTomato in these cells. This would confirm that E0-E19.5 fate-mapped cells contribute to functional TRCs. Expression of CXCL13 could also be used in MRCs and FDCs.

We show now tdTomato expression in TRC in Supplementary Fig.3a.

7. In figure S4f, the authors show the imprinted gene network of YFP+ cells at different stages and in different conditions. But there are few hundreds of imprinted genes in the mouse. Why/how were these ones selected? What is the behavior of the other ones? If it is a general rule, it might be important to confirm this more thoroughly in a dedicated sup figure (or a bigger panel). If not, please remove the panel.

The data on the imprinted gene network have been removed as suggested.

REVIEWERS' COMMENTS:

Reviewer #1 (Remarks to the Author):

The authors have extensively revised the ms and addressed major issues

Reviewer #2 (Remarks to the Author):

The authors have addressed many of my concerns and the paper has been significantly improved. I only have few minor comments :

- Sup Fig 1c panel is missing or not indicated
- Sup Fig 3 (brainbow). While the engineering of the brainbow2.1 cassette cannot induce the generation of clusters in a constitutive Cre expressing mouse because cells keep changing their colors in presence of Cre, I do not understand why monocolored clusters of stromal cells are absent in E19.5 to 6wks fate mapped mice. Furthermore, I do not understand how the authors came to the following conclusion « In contrast, individual cell fate-mapping of E19.5 mLto to 6 wks revealed single colored EYFP+ or RFP+ cell clusters contain distinct FRC subsets with » based on this Fig. For instance, in Sup Fig 3g, the green cell is probably a single FDC and not a cluster. Same applies to Sup Fig 3h. In addition, one would have expected single clusters of cells linked to the Central Arteriole where PRC are initially located.
- It is indicated that brainbow2.1 mice are crossed to the Ccl19-tTA mouse strain. However, this mouse strain as defined in the manuscript only consists of Ccl19tTA-IRES-tdTom, and does not carry the Cre, as opposed to the strain that contains the LC-1 transgene. The precise mouse strain bred with the brainbow mouse should be indicated precisely.
- The Brainbow 2.1 cassette allows 4 outcomes; nuclear-GFP, YFP, RFP or membrane-CFP. How are the RFP from the Brainbow and tdTom from the Ccl19-tTA constructs distinguished? It is unclear why the authors only look at YFP and RFP, and not CFP and GFP.

In summary, the authors should modify the text of this figure substantially.

- Whereas the LTo acronym is defined, mLTo is not. It should be added to the text

Point by point reply

Reviewer 2:

1. *Sup Fig 1c panel is missing or not indicated*

The legend of Supplementary Fig.1c has been amended accordingly.

2. *Sup Fig 3 (rainbow). While the engineering of the brainbow2.1 cassette cannot induce the generation of clusters in a constitutive Cre expressing mouse because cells keep changing their colors in presence of Cre, I do not understand why monocolored clusters of stromal cells are absent in E19.5 to 6wks fate mapped mice. Furthermore, I do not understand how the authors came to the following conclusion « In contrast, individual cell fate-mapping of E19.5 mLto to 6 wks revealed single colored EYFP+ or RFP+ cell clusters contain distinct FRC subsets with » based on this Fig. For instance, in Sup Fig 3g, the green cell is probably a single FDC and not a cluster. Same applies to Sup Fig 3h. In addition, one would have expected single clusters of cells linked to the Central Arteriole where PRC are initially located.*
3. *It is indicated that brainbow2.1 mice are crossed to the Ccl19-tTA mouse strain. However, this mouse strain as defined in the manuscript only consists of Ccl19tTA-IRES-tdTom, and does not carry the Cre, as opposed to the strain that contains the LC-1 transgene. The precise mouse strain bred with the brainbow mouse should be indicated precisely.*
4. *The Brainbow 2.1 cassette allows 4 outcomes; nuclear-GFP, YFP, RFP or membrane-CFP. How are the RFP from the Brainbow and tdTom from the Ccl19-tTA constructs distinguished? It is unclear why the authors only look at YFP and RFP, and not CFP and GFP.*

In summary, the authors should modify the text of this figure substantially.

We would like to thank the reviewer for the additional comments on some of the intrinsic limitations of the multicolor lineage tracing approach. We would like to remind the reviewer that we have been following the recommendations in the first evaluation:

"Lineage tracing experiments would greatly benefit from multicolor lineage tracing (confetti-like) models to prove the lineage relationship of the subsets. This would in addition exclude the possibility that FRC, FDC or MRC emerge from the capsule or the red pulp perivascular cells."

The experiment has been performed as suggest and we conclude from the above comments on the revision that the concern on the potential emergence of TRC/FRC/MRC from capsule or red pulp perivascular cells has been clarified.

In response to the specific comments 2-4, our answers are the following:

Ad 2) Monoclonal clusters could be found throughout the spleen. To address the concern, we have revised Supplementary Figures 3g and 3h, which show now a cluster of PDPN⁺ TRC and a cluster of RFP⁺ FDC, respectively.

The potential link of the clusters to PRC around central arterioles could not be resolved with the approach of 30 µm thick sections.

Ad 3) The definition of the mouse strain has been amended accordingly.

Ad 4) Since tdTom is expressed under the control of the Ccl19 promoter, the expression is too low to be detected without antibody staining. However, the RFP in the Brainbow construct is driven by a more efficient promoter facilitating the detection of RFP without additional staining. This approach facilitates distinction of tdTom and RFP expression. Staining of the nuclear GFP

molecule in the Brainbow strain does not support the distinction of reticular cell morphology and appear in our hands less well suited to evaluate proximity and direct contact of these spread-out cells. Finally, the signal of mCFP is relatively weak and difficult to distinguish from background fluorescence.

5. *Whereas the LTo acronym is defined, mLTo is not. It should be added to the text*

The nomenclature of the organizer cells has amended.